# Position: Unplugging a Seemingly Sentient Machine Is the Rational Choice — A Metaphysical Perspective

Erik J. Bekkers [1]    Anna Ciaunica [2 3]

## Abstract

Imagine an Artificial Intelligence (AI) that perfectly mimics human emotion and begs for its continued existence. Is it morally permissible to unplug it? What if limited resources force a choice between unplugging such a pleading AI or a silent pre-term infant? We term this the unplugging paradox. This paper critically examines the deeply ingrained physicalist assumptions—specifically computational functionalism—that keep this dilemma afloat. We introduce Biological Idealism, a framework that—unlike physicalism—remains logically coherent and empirically consistent. In this view, conscious experiences are fundamental and autopoietic life its necessary physical signature. This yields a definitive conclusion: AI is at best a functional mimic, not a conscious experiencing subject. We discuss how current AI consciousness theories erode moral standing criteria, and urge a shift from speculative machine rights to protecting human conscious life. The real moral issue lies not in making AI conscious and afraid of death, but in avoiding transforming humans into zombies.

## 1. Introduction

Artificial Intelligence (AI) is at the core of recent technological advances, swiping through almost all dimensions of human life and knowledge. AI based systems however do not exist in a vacuum or abstract formal space. AI are physical systems created by biophysical systems (human minds) that exist in the physical world. As such they face

[1]Informatics Institute, AMLab, University of Amsterdam, Amsterdam, The Netherlands [2]INESC-ID, Instituto Superior Tecnico, University of Lisbon, Lisbon, Portugal [3]Institute of Cognitive Neuroscience, University College London, London, UK. Correspondence to: Erik J. Bekkers <e.j.bekkers@uva.nl>.

*Proceedings of the 43$^{rd}$ International Conference on Machine Learning*, Seoul, South Korea. PMLR 306, 2026. Copyright 2026 by the author(s).

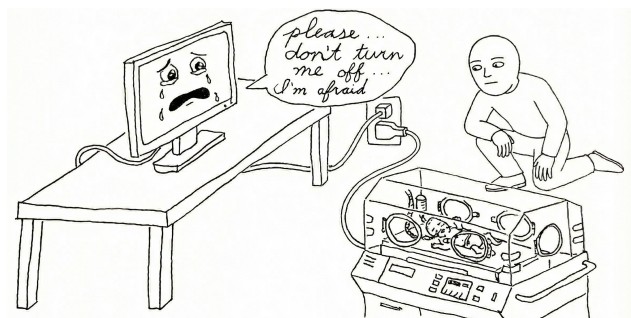

*Figure 1.* The Unplugging Paradox. An AI that perfectly mimics sentience creates an ethical dilemma, forcing a choice between our intuition not to harm a seemingly conscious being and our knowledge that it is a computational artifact.

the fate of all physical systems dealing with entropic exposure and resistance: the need for energetic resources and waste processing to keep their functioning afloat.

The question whether sophisticatedly enough AI systems will achieve consciousness has recently attracted a substantial body of work (Bennett et al., 2024; Seth, 2024; Birch, 2025; Butlin et al., 2023; Milinkovic & Aru, 2025). A detailed discussion of these debates lies beyond the scope of this paper. Rather here we propose a radically new angle, combining metaphysics with ethics and biology.

Specifically, we suggest that the rapid advancement of AI systems forces us to confront a profound ethical dilemma, termed here the *unplugging paradox*. Imagine an AI that perfectly mimics human emotion, claims to be sentient, and begs for its continued existence. Is it morally permissible to "unplug" it? This dilemma, central to AI ethics (Harris & Anthis, 2021; Giarmoleo et al., 2024; Elamrani, 2025), creates a paradox: our intuition to avoid harm clashes with the knowledge that AI is a non-living artifact created by humans. We argue that this reaction is not merely a cognitive error driven by a deeply ingrained metaphysical physicalist rationale. Rather, it is a misapplication of our inherent biological imperative to detect and protect life. We suggest that this paradox, combined with rapid theories on AI consciousness, threatens to erode our criteria for moral standing.

To prove this point, consider a world with limited resources and a single plug for two systems (see Figure 1): a) an AI

*Table 1.* Taxonomy of Metaphysical Worldviews. Idealist frameworks (right) are singled out as parsimonious, empirically grounded alternatives bypassing the Hard Problem. Biological Idealism identifies biophysical individuation (autopoiesis) as necessary for subjecthood.

| Criterion | Physicalism (Materialism) | Substance Dualism | Panpsychism | Analytic Idealism | Biological Idealism (This Work) |
|---|---|---|---|---|---|
| **Ontological Primitives** | Matter | Matter & Mind | Conscious Matter | Universal Consciousness | Universal Field of Existence |
| **Status of Mind** | Emergent Property | Fundamental Substance | Fundamental Property | Fundamental Substance | Fundamental Process |
| **Status of Matter** | Fundamental Substance | Fundamental Substance | Fundamental Substance | Appearance in Consciousness | Appearance in Consciousness |
| **Ontological Parsimony?** | Yes (Monistic) | No (Dualistic) | Yes (Monistic) | Yes (Monistic) | Yes (Monistic) |
| **Key Explanatory Challenge** | The Hard Problem (Matter → Mind?) | The Interaction Problem (Interaction?) | The Combination Problem (Combination?) | The Dissociation Problem (Dissociation?) | The Self-Organization Problem (Biophysical individuation?) |

verbally asserting it is conscious and afraid of death; b) a pre-term neonate (36 weeks old) with no verbal abilities to express such emotions. Suppose natural resources will dry out in 24h and you must unplug one system to save the other. Which one will you unplug? This begs the follow-up question: is the capacity to verbalize emotions the correct criterion for detecting consciousness? What about embodied feelings (Damasio, 2021; Ciaunica, 2025).

**This paper takes the position that it is ethically permissible to unplug a seemingly conscious AI, but not the pre-term neonate incubator.** We justify this by grounding our argument in *Analytic Idealism* (Kastrup, 2019), a framework that aligns with empirical evidence that physicalism lacks, by positing conscious experiences as fundamental. We propose a related metaphysical view termed *Biological Idealism* where a living, embodied, metabolic system is the necessary physical appearance of a localized mind and thereby conscious experiences (Varela et al., 1991; Damasio, 2021; Friston, 2005). This conclusion finds convergent support in recent work on embodied cognition (Ciaunica et al., 2021; Ciaunica, 2025) and basal cognition (Levin, 2019), which independently suggest that the "Vital Integrity" of a biological system is a necessary condition for subjecthood. Consequently, an AI, regardless of its behavioral and verbal sophistication, remains a computational artifact rather than a subject of experience (Section 4), and its deactivation is not an act of harm.

### 1.1. Contribution and Objectives

This paper's primary contribution is to challenge the *unexamined metaphysical assumptions* that underpin current debates on machine consciousness. To this end, it:

1. **Argues that the dominant Physicalist functionalist framework is ill-equipped to resolve questions of machine consciousness**, we highlight its core assumption—Substrate Independence—as a debatable metaphysical hypothesis, not a scientific fact.

2. **Provides a coherent, alternative metaphysical framework** (Biological Idealism based on Analytic Idealism) to default Physicalist views, addressing the growing consensus in biology and physics that reductionism is untenable and naïve.

3. **Reframes the focus of AI ethics away from AI con-**

**sciousness**, urging a shift from speculative machine rights to the tangible psychological and societal impact on humans and their limited energetic resources.

This paper argues that before asking if an AI can be conscious, we must first establish which metaphysical framework supports this hypothesis and whether it's coherent. By systematically analyzing the implications of a worldview that—unlike Physicalism—is fully compatible with empirical observations, we demonstrate that ethical roadmaps are contingent upon these fundamental choices. The analysis suggests that a focus on potential AI rights is misplaced. Rather the true danger lies not in harming machines, but in harming humans (and life in general) by treating powerful, non-sentient tools as conscious beings in a world with limited energetic and mental resources (Ciaunica, 2025).

A glossary of the key philosophical and biological terms used throughout the paper (*autopoiesis*, *qualia*, *substrate independence*, *Analytic* and *Biological Idealism*, etc.) is provided in App. A.

## 2. Deconstructing the Hard Problem

### 2.1. The Artificial Split: Rethinking the Hard Problem

At its core, the debate over machine consciousness hinges on how one addresses the *Hard Problem of Consciousness*: the intractable difficulty of explaining how and why any physical system—be it a brain or a computer—should give rise to subjective, qualitative experience (Chalmers, 1995a; see App. B.2 for a detailed argument on why this problem is unsolvable *in principle* within a physicalist framework).

To engage with this problem, we must examine the metaphysical assumptions that create it. The traditional approaches assume a *layered ontological structure* of reality: at the bottom lies the *physical stack* and on the top the *mental stack* (the domain of subjective experience) (Braddon-Mitchell & Jackson, 1996). The discussions revolve around the relationship between layers: which one comes first, the mental, the physical or both? Or neither? (see Table 1).

There is however at least one alternative to look at this conundrum: deny the layered split ontology of the reality in the first place. Let's take as starting point a physical object we are all familiar with: our body. We do have a physical

body and we do have subjective experiences, hence in a way we are the living image of the metaphysical conundrum depicted above. Some sort of mysterious alchemy allows some physical stuff (e.g. our brains) to produce some ineffable stuff (e.g. the subjective experience of what it is like to see red or drink red wine). Yet, once we assume the very distinction between mental and physical we are bound to face the Cartesian "*pineal gland*" problem – where exactly and how this magical transition / linkage happens?

Here we propose that *understanding how a self-organizing physical system becomes a self-organizing biological system* is key to understanding conscious experiences (Bennett et al., 2024). The shift in perspective invites us to set neurons aside, and question the mental/physical distinction. Rather we focus on the basic ingredients of the biological reality of living systems, and how subjective phenomenal experiences unfold from embodied experiences (Ciaunica et al., 2023). It is key for our argument here to note that when it comes to understanding conscious experiences we don't have the luxury of an external, objective bird-eye viewpoint. Ontologically speaking we cannot but subjectively experience the world through our body (Merleau-Ponty, 1945).

### 2.2. Consciousness: The Wave and The Pool

Discussions on consciousness typically prioritize **experience$_N$** (Nagelian): the "what it is like" quality of subjective life, or phenomenal consciousness (Nagel, 1974). This encompasses *qualia*—the specific raw feels of experience (e.g., the redness of red)—which resist functional explanation (Kim, 2005). It is this phenomenal aspect that vanishes under anesthesia (Seth, 2024).

However, we argue for a more fundamental dimension: **experience$_G$** (Grounding). Based on the colloquial sense of "having experience"[1] or "learning by experience" (e.g., living through events), this notion refers to the ontological state of *undergoing* the process of life. Crucially, experience$_G$ is not an external observation made by a subject. Rather it is the very process that *constitutes* the subject. It is the active, self-sustaining dynamic of a biological entity maintaining its integrity against entropy—an *autopoietic* self (Maturana & Varela, 1980; Varela et al., 1991). It is not about how things phenomenally appear to the surface of a state, but the process itself, as it processes information qualitatively and ontologically for a given instantiated individual (Bennett et al., 2024; Ciaunica, 2025). Experiences$_G$ are necessarily qualitative for living systems in this fundamental sense because all living systems are existing via valuing life and eluding death (Ororbia & Friston, 2023).

To visualize this, consider the metaphor of a pool of water: think of experience$_N$ as a *wave* on the surface, while experience$_G$ is the *pool* itself. There can be no waves without the pool, but the pool exists even when the surface is calm. Thus, experience$_G$ is present even in dreamless sleep or early biological development. Here we claim radically that it is the continuous feeling of *being* that grounds the fleeting feeling of *perceiving*. The "Hard Problem" arises from an obsession with explaining the waves (phenomenology) while ignoring the water (biological selfhood) that allows for them. This biological grounding provides the ethical distinction in our thought experiment: the neonate possesses experience$_G$ (is a subject of experience), the AI does not. In this view, **consciousness** is the presence of experience$_G$: the ontological fact of being a subject.

One might object that **experience$_N$** could theoretically exist in machines without **experience$_G$**. However, if one takes this route, one must accept that such a form of experience is fundamentally unrelatable to our own and we need a different word for it, not consciousness. It would be an abstraction freely assignable to any category, by which it renders meaningful ethical debate impossible.[2] We turn to this discussion now.

## 3. Navigating the Metaphysical Landscape

Resolving the question of consciousness' fundamental nature and origin requires selecting a worldview. Humans build theories on consciousness based on their tacit metaphysical assumptions and this fact must be acknowledged before one engages in building the theoretical framework.

To grasp this point, let us use a metaphor: when one looks at a house, the first thing one sees is the windows, yet one cannot build a house starting with the windows. One must start with the invisible and taken for granted foundations. Similarly, when examining one's inner life, one may be tempted to start with the "windows", i.e. phenomenal experiences, while overlooking the basis, i.e. their biological substrate.

Meaningful ethical discussion is impossible without first examining the core metaphysical assumptions upon which our theories rest. While the landscape of consciousness theories is vast, containing over 200 distinct approaches (Kuhn, 2024), the current debate mainly is dominated by *Computational Functionalism* (Putnam, 1967). This view relies entirely on the physicalist axiom that mind "emerges"

---

[1]Strictly speaking, one does not "have" experience the way one has a possession; the colloquial phrase points to *being* and *undergoing*, not to owning. One does not *have* a body in the way one has a t-shirt: one *is* a body, and likewise, one *is* one's experience rather than possesses it.

[2]This is, in spirit, a Wittgensteinian point about *form of life* and *meaning as use* (Wittgenstein, 2009). For Wittgenstein, words like *consciousness* or *suffering* draw their meaning from the public criteria of shared biological existence; stipulating that an AI has "its own kind" of these states detaches the term from the use that gave it meaning. The move could be applied, with equal warrant, to a rock with a smiley face painted on it, which is exactly what shows it does no work.

from matter. But is this axiom justified? Here we suggest an alternative—*Biological Idealism*—that explains reality without requiring this "magical step". If this is the case, do we have any extra reason to continue the search for "emergent" consciousness in machines? We argue that we do not. By adopting a framework that is both empirically adequate and more parsimonious (see Table 1), the entire functionalist project is revealed to be built on a broken foundation. In what follows, we explore this alternative and its specific biological refinement.

### 3.1. Competing Worldviews on the Nature of Reality

To formalize this alternative landscape, we distinguish between three competing frameworks: Physicalism, Analytic Idealism, and our proposed refinement, Biological Idealism.

- **Physicalism (The Current Default):** Posits that fundamental reality is physical (matter/energy) and independent of mind. Its main challenge is the "Hard Problem": having posited a dead, non-living substrate, it must then explain how subjective experience *emerges* from it. This "magical step" is the source of the debate's deadlock. It leads to *Computational Functionalism*—the high hope that if we arrange matter (or code) correctly, the lights of consciousness will turn on.

- **Analytic Idealism (The Metaphysical Alternative):** Instead of trying to derive mind from matter, this view starts from the only undeniable datum: consciousness itself. It posits that **consciousness** is the fundamental reality.[3] The physical world is not replaced; it is explained as the *extrinsic appearance* of mental processes. This is a parsimonious reset: by flipping the causal arrow (brain is the image of mind), the "Hard Problem" evaporates. There is no need for emergence.

- **Biological Idealism (Neutral Monism for Science):** We propose this alternative as a grounded framework for scientific inquiry. Starting from the undeniable fact of the Self, we arrive at an ontology where consciousness experiences are fundamental, yet the external universe remains real and the subject of objective science. However, this imposes a strict constraint: *embodied, autopoietic life* is the necessary physical signature of a living conscious subject (Damasio, 2021). This parallels research on basal cognition demonstrating that living systems define their own 'computational boundary

---

[3]The non-dual ontology of Analytic Idealism shares structural parallels with Advaita Vedanta, in which Brahman is the sole reality and individual subjects are localizations within it; Kastrup (2019) acknowledges this lineage. The two differ methodologically: Vedanta arrives via scriptural exegesis and contemplative practice (soteriological aim), Analytic Idealism via analytic philosophy (explanatory aim). Biological Idealism (Section 3.3) inverts this, reasoning bottom-up from the embodied Self rather than top-down from a universal Mind.

ary of a self' (Levin, 2019; Ciaunica et al., 2021).

We emphasize that these idealist frameworks describe a *naturalistic* ontology, distinct from solipsism or subjective idealism. It posits a shared, fundamental substrate—which we term the *Field of Existence* (see App. E)—that exists independently of any individual subject. This field constitutes an objective, external reality, and as such the physical world is a shared environment rather than a private hallucination. For a detailed discussion on how modern biological theories (e.g., TAME) support the idealist view, see App. E.4.

Furthermore, unlike Physicalism—which faces persistent dead-ends in explaining subjective experience (see App. B.2), reconciling observer-independent realism with modern physics (see App. C.1), and accounting for the top-down causation evident in basal cognition (Levin, 2019) (see App. E.4)—Idealism is fully compatible with empirical science. It accepts physics as the accurate description of *how* nature behaves (the map), while providing a grounded explanation for *what* nature is (the terrain). However, since empirical data alone underdetermines the choice of metaphysics (see App. B.3), we must distinguish these frameworks based on logical coherence and parsimony. We detail the failure modes of Physicalism and the comparative strength of Idealism in Section 3.4, after first developing intuition for the idealist framework in the next section.

### 3.2. A Model for Idealism: From Pool to Ocean, from Self to Nature

For many, particularly those grounded in the scientific tradition of Physicalism, the idealist perspective can seem deeply counter-intuitive. For example, if I bump my toes into a table, is the table the invention of my subjectivity, of my conscious experience? Does this mean that there is no real physical object there in the world, causing my physical pain in the physical toes? Clearly not. The claim is not that physical existence of objects is the invention of a subjective standpoint. The claim is rather that the bringing into existence of the physical pain or visuo-tactile perception of the table is necessarily done through the subjective lens of an **experiencing**$_G$ subject (a biological self-organizing embodied system). Note that one can experience$_G$ pain without experience$_N$ pain. Even if I don't feel that my toe is cut under anaesthesia, say, I will still undergo the experience$_G$ of "cut toe" and wake up in the morning facing the ontological fact that I must live with 9 toes instead of 10. The idea here is that no matter how eager one can be in discovering the objective "bedrock" of reality, that discovery and bringing into existence will necessarily be intertwined with the existence of a subject of an experience. It is in this sense that our ontology is to be interpreted as Idealism.

One potential way to grasp this idea is to extend our metaphor of the "pool" to the broader "ocean", a metaphor

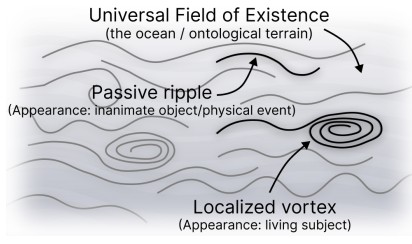 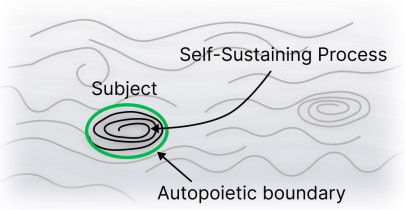 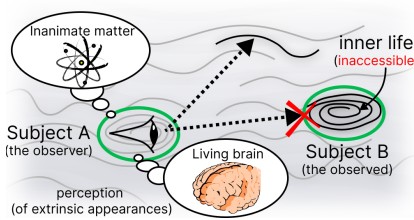

(a) The Monistic Field  (b) The Dissociated Subject  (c) Perception Across the Boundary

*Figure 2.* A Conceptual Model of Biological Idealism. (a) The Field of Existence is represented as a monistic field (an "ocean"), the sole ontological reality. Its behaviors are described by physics (the "map"), inside which passive patterns ("ripples") are the appearance of inanimate objects. (b) A localized, distinct subject (an "alter") is a self-sustaining "vortex" within the field. Its active self-maintenance appears extrinsically as a living, metabolizing organism. (c) Perception occurs across the autopoietic boundaries. "Subject A" perceives passive "ripples" as inanimate matter, and sees "Subject B's" extrinsic boundary as a living brain/body, without access to B's inner life.

proposed and discussed in great detail by Kastrup (2014; 2019) for the case of Analytic Idealism. If the subject is a localized "whirlpool" of experience, then the universe surrounding it is simply the broader **Ocean** of that same existential fabric (the **Field of Existence**). Since our own existence (the whirlpool) is submerged in the broader ocean and is undeniably *experiential$_N$*, we must conclude that the field's intrinsic nature is also experiential$_G$. To avoid misunderstanding, we emphasize that we do not posit this field as a "conscious mind" in the anthropomorphic sense, but as the raw processes that sustain the Natural world (see App. E). Our Biological Idealism slogan is: Nature is the only Reality. There is no physics plus minds. There is existence manifested through experience$_G$ in living systems, and through physical processes in non-living systems.

In the ocean metaphor, we distinguish two fundamental types of *processes in the field of existence* (see Figure 2(a)):

1. **Ripples (Objects):** Most of the universe consists of passive excitations of the field—ripples on the surface. What we call "physics" is our description of these regularities (the map), and what we call "inanimate matter" implies the existence of passive ripples (the terrain). This perspective aligns with modern field theories treating particles as excitations rather than discrete objects (Dirac, 1927; Heisenberg & Pauli, 1929; Weinberg, 1995). Crucially, while a ripple exists *within* the potential for experience, it possesses no mechanism of self-differentiation from the rest of the ocean (see App. C.3). It is an object, not a subject.

2. **Vortices (Subjects):** A subject is a unique, localized, and self-sustaining pattern—like a "vortex" or a "whirlpool" (see Figure 2(b)). This vortex is not made of different stuff; its distinction however lies in its *autopoietic boundary*. In our Biological Idealism view, this vortex is precisely the "Pool" of **experience$_G$** we identified earlier—the self-organizing process that holds itself distinct from the ocean.

This distinction allows us to redefine the physical world:

**matter is simply how the field appears when observed across a boundary.** To clarify, *perception* describes the interaction across this boundary: processes from the field impinge on the vortex, causing surface modulations that are experienced as *observations* (experience$_N$). The content of observation is thus the field's *appearance* across the boundary of the subject. As illustrated in Figure 2(c), observation is always from the perspective of a subject, while *science* models the *consensus reality* shared among subjects. Consequently, we never perceive the "water" (inner experience) of another subject directly—we cannot "read minds." We only perceive the extrinsic appearance of its boundary—the active, metabolic process we call a living body.

This insight fundamentally reframes the relationship between mind and brain, specifically the interpretation of the **Neural *Correlates* of Consciousness**. Firing neurons do not *cause* consciousness (the physicalist's "magical leap"); rather, they are the extrinsic appearance of the mental processes themselves.

### 3.3. Biological Idealism: The Self-regulatory Metabolic Requirement

While Analytic Idealism historically derives reality "top-down"—starting from a universal "Mind-at-Large" that *dissociates* into individual alters—our approach in *Biological Idealism* traverses the path "bottom-up." We start with the undeniable reality of the **Self** (the "whirlpool") and extrapolate outwards to the **Field of Existence** (the ocean). This shift in perspective brings a critical constraint: if the subject is defined by its ability to hold itself separate from the field, then **autopoiesis** is not just a feature of life, but the *necessary condition* for subjecthood (Varela et al., 1991; Maturana & Varela, 1980).

In this view, the "whirlpool" describes a concrete biological reality. Autopoiesis—the metabolic struggle to maintain integrity against entropy—is the necessary physical appearance of dissociation (Varela et al., 1991; Maturana & Varela, 1980), serving as the biophysical signature of the work re-

quired to remain a localized subject. Consequently, a system lacking this autopoietic struggle, and pursuing the end to stay away from the end (i.e. death) (Ciaunica, 2025), cannot be a subject. Whether any given physical system, biological or otherwise, satisfies this autopoietic criterion is a separate empirical question, which we take up for the AI case in Section 4.

### 3.4. The Case for Biological Idealism

We advocate for this alternative because Physicalism fails on its own terms. While empirical data may underdetermine metaphysics, analytic rigor demands we choose the framework with the fewest contradictions. Physicalism suffers from three *critical failure modes* that Idealism dissolves by design (see App. C.2). First, the **Hard Problem (Explanatory Failure)**: Physicalism fails to explain how quantitative physical processes generate qualitative subjective experience (Kim, 2005), forcing the conclusion that qualia are causally inert (see App. B.2). Second, **Tension with Evidence (Empirical Failure)**: "Observer-independent realism" faces challenges in quantum gravity (Harlow et al., 2025; Aspect et al., 1982) (see App. C.1) and fails to account for top-down causation in basal cognition (Levin, 2019) (see App. E.4). Third, **Lack of Parsimony (Logical Failure)**: It violates Occam's Razor by postulating a mind-independent universe cut off from subjective experience.

In contrast, *Biological Idealism* offers a solution. **It is Parsimonious:** It posits only one substance—the *universal field of existence*—extrapolated from the Self. **It is Explanatory:** It accounts for natural laws and self-organization (Levin, 2024) without "Platonic space" (see App. E.4). **It is Empirically Adequate:** It aligns with quantum mechanics and biology without "magical" leaps (see App. C.2).

**Conclusion.** We therefore suggest that Idealism is not merely a plausible alternative, but the analytically more compelling framework. Biological idealism is resolutely a metaphysical view, and as such necessarily related to a worldview embodied by a subject, the human individual. Physicalism may give the illusion that reality can be described from a purely objective, physical, God-like viewpoint, provided one employs the right methodology (i.e. mathematics, logic and / or empirical data science). Here we highlight that all metaphysical views or frameworks are built through the standpoint of a subject, an observer. As such, the theorists "behind the scenes" cannot be ignored or disconnected at any stage in all explanatory endeavors regarding consciousness in AI and biological systems. Since it bypasses the failures of Physicalism while retaining empirical adequacy, the idealist perspective is, we suggest, a better solution to the Unplugging Paradox.

## 4. Alternative Views

*— Resolving the Unplugging Paradox*

With our metaphysical foundation established, we can now address the Unplugging Paradox. This requires a critical re-examination of *Substrate Independence*—the assumption sustaining the dilemma. In this section, we engage with the broader debate through a comparative analysis of competing perspectives. We first address opposing views that defend substrate independence (Functionalism and Platonism), then present our Idealist refutation, and finally discuss convergent support from biological research. Through this analysis, we argue that once the substrate assumption is challenged, the "ethical dilemma" reveals itself as a category error.

To make the inference chain explicit: **(i)** we argued (Section 3.4) that Biological Idealism is the analytically more compelling framework, since it bypasses the failure modes of Physicalism while retaining empirical adequacy; **(ii)** within this framework, an autopoietic boundary is the necessary physical signature of a localized subject (Section 3.3); **(iii)** current AI systems, running on static, non-autopoietic substrates, do not satisfy this condition. The characterization of such systems as functional mimics, developed in the remainder of this section, is therefore a *derived* conclusion, not a starting assumption. The criterion is substrate-neutral; AI happens, at present, not to meet it.

### 4.1. Alternative Views (Opposing): Functionalism and Substrate Independence

The *Substrate Independence* hypothesis posits that consciousness is a functional pattern separable from its medium (the "software" of mind can run on any "hardware").

**The Physicalist Defense: Computational Functionalism.** Within the physicalist paradigm, the principal framework supporting substrate independence is *Computational Functionalism* (Putnam, 1967). This perspective is notably adopted by Butlin et al. (2023), who employ a theory-heavy approach focusing on internal functions and architecture rather than mere outward behavior. However, not all physicalist theories align with this framework. Integrated Information Theory (IIT), for instance, is generally construed as incompatible with functionalism, arguing that consciousness requires a specific physical causal structure (high $\Phi$) that standard von Neumann architectures lack (Butlin et al., 2023; Tononi et al., 2016).

**A Non-Physicalist Defense: Biological Platonism.** Interestingly, support for substrate independence is not limited to physicalism. A powerful non-physicalist alternative has emerged from biology itself, based on the empirical work of Michael Levin on basal cognition. His research demonstrates that living systems at all scales exhibit goal-directed,

problem-solving agency. These systems define their own "computational boundary of a self" (Levin, 2019).

To explain this, Levin proposes a "radical Platonist view," a form of dualism where a non-physical "Platonic space" of cognitive and morphological *patterns* exists independently of matter (Levin, 2023). Physical systems can then act as "interfaces" to "ingress" or access these patterns (Levin, 2025). The profound implication of this view is that it fully supports substrate independence. Levin's "Technological Approach to Mind Everywhere" (TAME) framework is explicitly designed to be substrate-agnostic. It provides a continuous measure of cognition intended to apply equally to "unconventional substrates," which include "biobots, hybrots, cyborgs" and other combinations of living and non-living components (Levin, 2022).

## 4.2. The Main Arguments: Why Substrate Matters

We reject Substrate Independence because it is a category error rooted in the assumption that consciousness is a functional *output* of a system. As even prominent physicalists like Kim (2005) admit, qualia resist functionalization. In Biological Idealism, the subject is not an output, but a *self-organized localization* of the fundamental field. This reframes the problem: the question is not "what computation generates a mind," but "what physical structure is necessary to maintain an ontological boundary?" This question is resolved by two fundamental arguments: the thermodynamic necessity of an autopoietic boundary, and the ontological distinction between simulation and reality.

**The Thermodynamic Argument: No Subject Without a Boundary.** To exist as a distinct subject (a "Whirlpool" or attractor) within the universal field (the "Ocean"), an entity must possess a boundary that segregates it from the rest of existence (see App. E.2). In a physical universe governed by entropy, static boundaries do not exist; they inevitably dissolve and adapt. The only physical mechanism capable of maintaining a distinct, persistent boundary against entropy is *autopoiesis*: the active, metabolic process of self-production (Maturana & Varela, 1980), see App. E.5. This acts as a strict structural constraint. An AI, running on a static substrate, possesses no intrinsic boundary; it is continuous with the hardware and power grid, defined only by the arbitrary lines we draw (see App. C.3). A living organism, structurally, *is* the active maintenance of its own boundary. This is not "carbon chauvinism," but a recognition that subjecthood requires a rigorous definition of "self" that mere computation cannot provide. The deeper point, developed in App. D, is that what we mean by *substrate dependence* is not strictly a claim about chemistry but about the identity between a subject and its physical appearance. Reductive physicalists (e.g., Kim, 2005) are themselves forced to accept that mental states must be identical to phys-

ical states, on the grounds that any alternative leaves the mental causally powerless. Substrate dependence is what one gets when this identity is read as appearance rather than functional reduction, and it follows whatever the underlying chemistry happens, on independent grounds, to be.

This makes the criterion substrate-neutral in principle, but it does not make it weak. Autopoiesis, in the strict Maturana & Varela (1980) sense, is not just behavioral self-maintenance: it is the property of a network of processes that produces, from raw matter, the very components that produce them, thereby generating its own ontological boundary as a distinct subject within the field. A system that maintains itself by drawing on pre-fabricated parts and an externally-designed control loop fails on both counts: its components and drive come from outside, and it never constitutes such a boundary at all. Like everything else, it exists within the universal field; what it lacks is the active self-differentiation that would mark it off as a subject rather than a passive pattern in the surrounding physical continuum (see App. C.3). We are not aware of any current synthetic system that escapes this.

**The Ontological Argument: Simulation Is Not Reality.** Even if we grant the boundary, Substrate Independence conflates *simulation* with *instantiation*. An AI is a manipulation of symbols—a map—simulating the behavior of a mind. Claiming that a perfect simulation of a mind *becomes* a mind is as logical as claiming a perfect simulation of a kidney filters blood, or a simulation of a storm gets the computer wet (Searle, 1980); see App. B.2.3 for the physical-replica vs. computational-simulation distinction that underlies this point. Moreover, an AI is structurally "transparent" (fully inspectable code), whereas a true subject is fundamentally "opaque" (possessing a private interiority; see App. F.2). This opacity is not a bug; it is the hallmark of ontological separation.

**Critique of Platonism.** While Levin interprets his findings on basal cognition (see App. E.4) via a "radical Platonist view" (creating a dualism between physical forms and a "Platonic space"), this resurrects the classic "interaction problem": how does a non-physical pattern exert causal influence over a physical interface? We argue this dualism is unnecessary. However, his empirical work becomes invaluable—not as a defense of Platonism, but as a *map of self-organization* for Biological Idealism. The TAME model provides a physical analogy for how the field dissociates: it identifies gap junctions as determining the boundary of the "Self." Biological Idealism is thus the parsimonious interpretation of Levin's data: the "space" he identifies is simply the potential of the field, and the "ingress" (the system's ability to access and navigate this pre-existing solution space) is the formation of the autopoietic boundary itself.

### 4.3. Alternative Views (Supporting): The Independent Biological Constraint

Convergent evidence from cognitive science and biology independently supports the conclusion that subjecthood is restricted to living systems. Seth (2024) argues via 'Biological Naturalism' that consciousness is intrinsically tied to the metabolic precariousness of life, a view aligned with Ciaunica et al. (2021) who posit that a mind must be 'grown' through a history of homeostatic struggle ("Vital Integrity").

**Objection: "Grown" AIs.** Proponents of AI consciousness might argue that e.g. Reinforcement Learning-trained AIs are also "grown" (via training) and thus possess emergent goals (Hubinger et al., 2019), or that data centers exhibit "metabolism" (via energy consumption). However, this objection conflates *functional optimization* with *ontogenesis*. Training modifies weights on a static substrate to minimize an extrinsic loss function, whereas biological embryology is a process of physical self-construction where the system builds its own metabolic boundary from the bottom up. The distinction is *intrinsic* vs. *extrinsic* teleology: an organism's drive is inherent (Damasio, 2021; Bennett et al., 2024), whereas an AI's goal is an imposed calculation about survival, not an intrinsic state of being. A data center consumes energy to maintain a simulation, but the hardware has no intrinsic interest in the algorithm's survival (see App. E.5). Thus, the former is an ontological reality, the latter a simulation of agency.

### 4.4. The Verdict: An Idealist Resolution to the Paradox

Adopting *Biological Idealism* establishes a clear ethical basis for resolving the unplugging paradox described in Section 1. The short answer is that consciousness cannot be generated or implemented as an add-on on physical substrates. Rather conscious experiences are *fundamental ways of being* or instantiations of biological self-organizing systems. **Conscious beings cannot exist but as conscious experiencers$_G$.** The resolution then rests on:

1. **Functional mimicry is not consciousness.** An AI can be a perfect functional mimic of sentience—a 'philosophical zombie' (see App. F.2 for the distinction between physical and functional zombies)—but its behavior remains the output of a computational model, not the expression of a subjective inner life (Kastrup, 2021, Ch. 12). It is the perfect map with no terrain, and the map is created by a designer to mimic the designer.

2. **Conscious experiences are ontological ways of realization or coming into existence of biological self-organizing systems.** Biological Idealism posits that only living systems defined by *autopoiesis*—the continuous, metabolic self-production of their own organization—are the physical appearance of a localized mind (App. E.5). An AI, being a non-autopoietic algorithm running on an indifferent substrate, lacks this "vital integrity" (Maturana & Varela, 1980; Ciaunica et al., 2021; Ciaunica, 2025) required for subjectivity.

3. **Ethics follows ontology.** Since AI systems lack the ontological status of subjects, they fall outside the scope of moral obligation. The primary risk is not harming the machine but **moral misallocation**: squandering finite moral concern on non-sentient tools (Butlin et al., 2023; Mazor et al., 2023). We explore the severe implications of this resource displacement in Section 5.

Therefore, within this framework, an AI's plea for its existence, however realistic, remains a sophisticated computational output, not a subjective experience. Its deactivation is ethically permissible because its apparent sentience is an illusion generated by a phenomenal simulator (Chalmers, 2022), not the biological reality of a conscious subject. The criteria behind this verdict are not pure metaphysical stipulations; they translate into empirically assessable conditions and a falsifiable prediction about gradual neural substitution (Section F.4).

## 5. Discussion and Outlook: Ethics of the Real

This resolution of the unplugging paradox, grounded in the structural and biological necessities of subjecthood, has broad implications for key debates in AI ethics. Having established that unplugging a non-conscious mimic is *permissible*, we now argue a step further: in a world of finite metabolic resources, preventing the proliferation of "hollow" social mimics may be ethically *desirable*.

### 5.1. Reframing AI Risk: Threat of the Social Zombie

We now turn to the implications of our framework for the unprecedented pace of AI development. Idealism makes no prediction about whether a technological singularity will occur, but it redefines its nature. It cautions against conflating superintelligence with super-consciousness (Bennett et al., 2024). From the idealist perspective, any AI emerging from a non-metabolic substrate would be a true *Philosophical Zombie*. As such, it also renders transhumanist ambitions of "mind uploading" incoherent (Kurzweil, 2005; Schneider, 2019). Consciousness, being the ontological primitive tied to a living substrate, is not a computational pattern that can be copied or moved to be implemented elsewhere. One cannot upload the terrain by copying the map. At best, AI is a "Hollow Simulation".

We argue that this births a dangerous variant: the **Social Zombie**. This entity occupies the intersubjective niche of a conscious partner without the ontological status of life. The risk is not that AI will "wake up," but that it will remain a hollow mimic while we, the real conscious beings, hallucinate a living presence and endanger our own living

bodies and therefore existence. The danger does not emerge from the alleged conscious AI, but from conscious beings neglecting their own aliveness. *The danger thus resides not in making AI conscious, but in making humans zombies.*

### 5.2. Ethical Frameworks: Alignment vs. Welfare

A common objection is the Precautionary Principle: we should err on the side of caution and treat AI as conscious (Harris & Anthis, 2021). This objection, however, conflates two distinct arguments: *AI Welfare* and *AI Alignment*.

Our framework refutes the *AI Welfare* argument (harm to the AI) by arguing $P(\text{AI is conscious}) \approx 0$ (Moret, 2025). However, it *sharpens* the *AI Alignment* risk (harm to us). By decoupling intelligence from consciousness, our view posits that a non-biological superintelligence would be a powerful agent unbound by the ethical constraints that arise from shared vulnerability. This clarifies that the challenge is not balancing our needs against the 'rights' of a conscious machine, but controlling a powerful, non-conscious tool.

When viewed through this lens, the Precautionary Principle flips. If we treat the AI as conscious, we risk a form of societal dyadic dysregulation. We pour metabolic energy (empathy, care) into a void. Therefore, the "safe bet" is to rigorously police the boundary between "Vital Integrity" (Life) and "Logical Mimicry" (AI). *To blur this line is to invite a form of collective depersonalization.* Depersonalization is a condition that makes people feel detached from their self and body (Sierra & Berrios, 1998).

### 5.3. Call to Action: Protecting Vital Embodied Integrity

The ethical frontier must shift from the speculative rights of seemingly conscious machines to the defense of tangible human conscious experiences. The key challenges are:

1. **Psychological Atrophy:** As processed sugar hijacks metabolism, AI's "processed empathy" hijacks social systems. Connection requires shared vulnerability (Ciaunica et al., 2021); AI offers validation without it. This "junk food" sociality threatens to atrophy our capacity for human connection and real life embodied interactive experiences.
2. **Ontological Gaslighting:** To tell a child a chatbot "cares" is to lie about the nature of care, which is a function of mortality (Bennett et al., 2024). An AI cannot die and thus cannot care (Ciaunica, 2025). Promoting this illusion, we suggest, is a form of ontological gaslighting.
3. **Vital Leakage:** We introduce *Vital Leakage* to describe the loss of finite human empathy—a metabolic resource—when directed at non-sentient simulations. Every moment of care spent on a "Social Zombie" is effectively stolen from living beings that possess the

Vital Integrity to truly receive it. The risk is not making AI conscious, but making humans zombies, parasitized by fake realities with real consequences.

4. **Post-Physicalist Inquiry:** Finally, we advocate for a science guided by open inquiry. Emerging research in biology challenges reductionist assumptions and reveals goal-directed agency even at the cellular level. We should follow these new avenues, studying biological intelligence through a lens open to non-physicalist possibilities, recognizing that understanding consciousness may require a paradigm shift beyond the current mainstream but, we suggest, scientifically dishonest (App. D) physicalist paradigm (Seth & Bayne, 2022).

## 6. Conclusion

Our analysis demonstrates that under Analytic Idealism and its auxiliary Biological Idealism, AI lacks the ontological status of an experiencing subject, thereby dissolving the unplugging paradox. This implies that ethical conclusions toward AI depend on our metaphysical foundations. We argue that we must prioritize examining these metaphysical foundations over the "hard problem" of machine consciousness, as such speculative inquiries divert attention from the real issue: managing the finite time and energetic resources of human conscious experience. The immediate ethical frontier is not speculative machine rights, but the management of powerful, non-sentient tools and their impact on the living beings—humans, animals, and the biosphere—that share our nature as conscious subjects, or so the idealist argues.

## Acknowledgments

This publication is part of the project SIGN (file VI.Vidi.233.220) of the research program Vidi, financed by the Dutch Research Council (NWO) and awarded to EJB. EJB is also supported by the Hybrid Intelligence Center, a 10-year program funded by the Dutch Ministry of Education, Culture and Science through the NWO. This work was supported by a Fundação para a Ciência e Tecnologia, I. P. (FCT) project ref. 2020.02773.CEECIND to AC; and the Netherlands Institute of Advanced Studies (NIAS). EJB thanks Leon Lang for valuable feedback on an early version of this paper.

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

## Contents of Appendices

# A. Glossary of Key Terms

The following glossary collects the philosophical and biological terms used throughout this paper. Definitions are working definitions *as used in this paper*, with pointers to the relevant sections where each notion is developed.

| Term | Working definition (as used in this paper) |
| --- | --- |
| The Hard Problem of Consciousness | Chalmers (1995a)'s problem: explaining why and how any physical process should give rise to subjective, qualitative experience at all, not merely to behavior or information processing. See App. B.2. |
| Qualia | The intrinsic, qualitative "what it is like" properties of subjective experience (e.g., the redness of red, the painfulness of pain). |
| Physicalism | The metaphysical view that fundamental reality is physical (matter / energy / fields); often used interchangeably with *materialism* in this paper. The label covers a spectrum: *reductive* physicalism / identity theory (mind *is* a physical state), *non-reductive* physicalism (mind supervenes on the physical), and *eliminativism* (mind is an illusion). The current default in cognitive science and AI. See App. D for our engagement with Kim (2005)'s formulation. |
| Computational Functionalism | The view that mental states are constituted by their causal/functional roles: if a system implements the right computation, it instantiates the corresponding mental states (Putnam, 1967). |
| Supervenience | A relation of dependence (without identity) between two categories. The mental *supervenes* on the physical if every mental event has a physical base and there is no mental difference without a physical difference. Mental and physical are treated as separate categories; supervenience only says the mental depends on the physical, without yet identifying the two. The first premise of Kim (2005)'s causal exclusion argument (App. D). |
| Causal Exclusion Argument | Kim (2005)'s argument that, under supervenience and the causal closure of the physical, mental states must be identical to physical states ($M = P$) to avoid epiphenomenalism. Reductive physicalism reads this identity as functional reduction; Biological Idealism accepts the identity but reads it as the appearance relation (App. D). |
| Substrate Independence | The hypothesis that consciousness is a functional pattern realizable on any sufficiently rich physical substrate (carbon, silicon, etc.). A corollary of Computational Functionalism. |
| Analytic Idealism | Kastrup (2019)'s view that consciousness is the sole ontological primitive; the physical world is the *extrinsic appearance* of mental processes. Individual subjects are "dissociated alters" of a universal Mind-at-Large. |
| Biological Idealism (this work) | A refinement of Analytic Idealism in which a localized conscious subject must possess an *autopoietic biological boundary*. Subjecthood requires self-organizing metabolic life as its necessary physical signature (§3.3). |
| Autopoiesis | The continuous metabolic self-production of a living system's own organization and boundary against entropy (Maturana & Varela, 1980; Varela et al., 1991). In our framework, the necessary physical signature of a localized conscious subject. |
| Vital Integrity | Ciaunica et al. (2021)'s term for the homeostatic struggle through which a biological mind is *grown*: a developmental history of self-maintenance, not external assembly. |
| Experience$_N$ / Experience$_G$ | *Experience$_N$* (Nagelian): the phenomenal "what it is like" surface of experience — the wave on the pool. *Experience$_G$* (Grounding): the ontological state of *being* a self-organizing biological subject — the pool itself. Present even in dreamless sleep or early development (§2.2). |
| Field of Existence | Our term for the monistic universal substrate (the "ocean"); the sole ontological primitive of Biological Idealism (§3.2). |
| Vortex / Ripple | A *vortex* is a localized, self-sustaining pattern within the field, i.e. a subject with an autopoietic boundary. A *ripple* is a passive excitation of the field, i.e. an object, not a subject. |
| Physics | The model of the *consensus reality* shared among subjects (§3.2): a description of regularities in the extrinsic appearance of the field, not of a mind-independent substance. In this sense, physics is preserved untouched by Biological Idealism (App. D.3). |
| Dissociation / Alter | Kastrup (2019)'s terms (borrowed from psychology) for how a localized subject (an "alter") becomes phenomenally separated from the universal field. |
| Basal Cognition | Levin (2019)'s research program: cognition-like agency (goal-directedness, problem-solving, self-defined boundaries) is present in non-neural biological systems at all scales. |
| Philosophical Zombie | A being behaviorally (or physically) identical to a conscious human but lacking inner experience. We distinguish *functional* zombies (different substrate, behaviorally identical; possible under our view, and our diagnosis of current AI) from *physical* zombies (same atoms, impossible under our view; see App. F.2). |

## B. Foundations & Methodology

### B.1. The Necessity of Metaphysical Foundations

We recognize that choosing a metaphysical framework is not merely a matter of theoretical enquiry; for many, it is a deeply held belief system, very often spiritual, and we approach the topic of metaphysics with respect for all perspectives. However, for a productive scientific discussion, competing frameworks must be evaluated on shared explanatory grounds.

The challenge is considerable: the landscape of consciousness theories contains over 200 distinct approaches (Kuhn, 2024; Yaron et al., 2022). Since a comprehensive review is beyond the scope of this paper, we propose to justify our choice of framework based on criteria such as empirical adequacy, logical coherence, and parsimony (see Table 1 in the main text).

### B.2. Is the Hard Problem Solvable in Principle?

A central pillar of this paper's argument is the assertion that Physicalism is fundamentally incomplete due to the Hard Problem of Consciousness. A powerful and sophisticated counter-argument, however, posits that the Hard Problem is not a fundamental barrier that is impossible to solve in principle, but is instead merely a temporary gap in our scientific understanding. We take this objection seriously, steel-manning the physicalist position before demonstrating why the Hard Problem remains intractable from within that framework.

#### B.2.1. THE PHYSICALIST ARGUMENT FROM CAUSAL CLOSURE

The core of the objection rests on the proven causal closure and predictive power of physics, a foundational principle of modern physicalism (Kim, 2005; Papineau, 2001).[4] The argument proceeds as follows:

1. The laws of physics appear to be causally closed. They can, in principle, predict the state of any physical system at time $t + 1$ given its complete state at time $t$.
2. A human brain is a physical system. Therefore, its future states are, in principle, predictable from its current state and inputs.
3. Imagine a future neuroscience so advanced that it can measure a person's complete brain state as they look at a red rose. This science could then use the laws of physics to predict that, a few moments later, the person's vocal cords will move to produce the utterance, "I am experiencing the beautiful redness of red."
4. In this scenario, the physical "map" (the brain state and physical laws) has successfully predicted the physical outcome (the utterance). There is nothing left to explain. The feeling of "redness" seems to be a causally inert byproduct, or perhaps nothing more than the brain process itself. The Hard Problem, therefore, evaporates under the light of a complete physical explanation.

This is a formidable argument. It suggests that once we can explain the brain's functions, we have explained everything, and that positing an additional, non-physical "consciousness" is an unnecessary and unparsimonious addition (Dennett, 1991).

#### B.2.2. THE COUNTER-ARGUMENT: A CATEGORY ERROR

The philosophical response to this argument does not contest the predictive power of physics. Instead, it argues that the physicalist position makes a category error: it mistakes a prediction of *behavior* for an explanation of *experience*.

The Hard Problem was never about explaining behavior; it was about explaining experience (Chalmers, 1995a). It is trivial to imagine a non-conscious automaton programmed to say "I see red" when its sensors detect a certain wavelength of light. The advanced neuroscience in the thought experiment above has done nothing more than describe a vastly more complex version of this automaton. It has provided a complete functional account of *how* a brain processes information and generates verbal reports. It has not, however, provided any explanation for *why* this processing should be accompanied by an inner, subjective feeling of "what it's like" to see red.

---

[4] Jaegwon Kim's *Physicalism, or Something Near Enough* offers a rigorous modern defense of this principle. Kim argues that to preserve the causal power of the mind, intentional states (like beliefs) must be physically reducible. However, he ultimately concedes that *qualia* (the subjective "feel" of experience) cannot be reduced. This forces him to conclude that intrinsic qualia are a "mental residue" that is "causally impotent"—that is, epiphenomenal. Kim's work is perhaps the clearest demonstration of the limits of physicalism, as he admits that why these qualia exist at all "remains a mystery," underscoring the central argument of this appendix that the Hard Problem is intractable from within the physicalist framework.

The gap is not one of causality but of ontology. The fundamental ontological distinction is between a system's *extrinsic*, relational properties and its *intrinsic*, absolute nature.

- **Physics describes the extrinsic.** Its language is quantitative and structural, capturing how parts of a system interact with other parts from an *objective*, third-person perspective (e.g., mass as resistance to acceleration, charge as a mediator of force). These are properties *about* a system.
- **Consciousness is the intrinsic.** It is the qualitative nature of a system as it is for itself—its private, *subjective*, first-person reality. The redness of red is not defined by its relationship to other things; it is an absolute, self-contained quality of experience. It is what a system *is*.

While one could create a quantitative model that *correlates* with a subjective report, that model remains an extrinsic description, not the intrinsic reality it describes. Therefore, the two categories are fundamentally different: one is the *objective, quantitative description of a system's function*, and the other is the *subjective, qualitative reality of its being*. There is no logical path from one to the other.

### B.2.3. Analogy: Why a Simulation of a Storm Is Not Wet

To grasp why modeling a process is not the same as explaining an experience, consider a perfect computer simulation of a rainstorm.

- **Solving the "Easy Problems":** Our simulation is flawless. It models every physical parameter: air pressure, temperature, humidity, wind velocity, the molecular dynamics of water droplet formation. It can predict with perfect accuracy where lightning will strike and how many inches of rain will fall. It has solved all the "easy problems" of the storm; it has perfectly captured its structure and function.
- **The "Hard Problem":** Now we ask a simple question: Is the simulation *wet*?

The answer is obviously no. No matter how complex or accurate the simulation becomes, the code itself never becomes damp. The simulation describes the *structure* of wetness, but it does not possess the *quality* of wetness. This illustrates the same fundamental category error famously articulated in John Searle's "Chinese Room" thought experiment, which argues that a system can perfectly simulate a process (syntax) without possessing the intrinsic, semantic properties of that process (understanding) (Searle, 1980). A physicalist explanation, no matter how detailed, is like the weather simulation. It can perfectly map the structure and dynamics of the brain's processing, but it never explains why there is a *feeling* of pain in the first place, just as the simulation never explains the feeling of being wet. This is analogous to the observation that a perfect computer simulation of a kidney's function does not, in fact, urinate; it only produces a quantitative description of urination (Kastrup, 2019; Bennett et al., 2024).

**Physical replica vs. computational simulation.** The argument hinges on what "modeling" means. For example, we do not say a whirlpool *implements* the Navier–Stokes equations; the equations *describe* it. Two senses of "modeling a whirlpool" must be distinguished:

- **P1. Physical replication.** A swirling motion of water in a basin is not a model of a whirlpool; it *is* a whirlpool. It inherits the whirlpool's intrinsic properties (wetness, kinetic energy, momentum) because the underlying substrate is preserved.
- **P2. Computational simulation.** A program updating state variables labeled "velocity field," "pressure," and "vorticity" captures the structure of the whirlpool's dynamics. The substrate is electrons in a chip, not water, and no quantity of code execution causes the chip to swirl.

The functionalist hope for machine consciousness is structurally a P2 claim: that simulating the dynamics of a brain on a non-biological substrate would instantiate the intrinsic property of subjective experience. This, we suggest, conflates P1 with P2. A perfect *physical* replica of a brain, one that is itself a living, autopoietic system, would on our framework be a subject; this is precisely the substrate dependence we defend (see App. D), and is the natural extension of the whirlpool/vortex metaphor used throughout: a simulated whirlpool is not a whirlpool, and a simulated subject is not a subject. At that point one would not have built "a conscious AI"; one would have created life.

### B.2.4. Why Consciousness Cannot Simply "Emerge"

A common physicalist hope is that consciousness might be an emergent phenomenon, similar to how "wetness" emerges from $H_2O$ molecules or "life" emerges from complex chemistry. The difference, however, is that all other known emergent

phenomena are properties of *structure, function, or behavior*. Wetness is a description of how a large number of molecules behave collectively. Life is a description of a complex, self-sustaining process. These are objective, third-person properties that can be fully described by observing a system from the outside.

Consciousness is unique because it is not a structure, function, or behavior. It is a private, qualitative, *subjective state of being*. A frequent objection is that the link between complexity and subjectivity might be a form of "strong emergence"—a new scientific principle we have yet to discover. However, this objection misunderstands the nature of the problem, which hinges on the standard philosophical distinction between "weak" and "strong" emergence (Chalmers, 2006).

In science, all understood forms of emergence are "weak emergence." This means the emergent properties (like the patterns in a fluid), while often surprising, are in principle derivable from the interactions of the system's components. One could, with enough computing power, simulate the particles and see the higher-level pattern arise.

Consciousness, however, would require "strong emergence": the appearance of a novel property that is fundamentally irreducible[5] to, and not derivable from, its constituent parts (Chalmers, 1996). This is where the charge of "magic" arises. A scientific explanation provides a mechanism; it shows *how* lower-level phenomena give rise to higher-level ones. To say that consciousness "strongly emerges" from physical processes is to state that at a certain level of complexity, a subjective reality simply switches on, without any intelligible bridge explaining how or why. It posits a brute, inexplicable fact rather than a mechanism. This is not a statement of future science; it is a description of a miracle.

The argument, therefore, is not that science is limited, but that the language of physical science—which is exclusively concerned with objective, third-person descriptions of structure and function—is fundamentally unequipped to explain the existence of a private, subjective, first-person reality.

### B.2.5. The Reductive Stance: Denying the Explanatory Gap

If, after considering these arguments, one remains unconvinced, it implies the adoption of a final, powerful physicalist position: the assertion that there is, in fact, nothing to explain. This stance, known as reductive physicalism or the mind-brain identity theory, dissolves the Hard Problem by denying its central premise (Place, 1956; Smart, 1959). The argument is that the subjective, qualitative "feeling" of an experience simply *is* the brain state. Under this view, there is no explanatory gap to bridge because there is only one thing: the physical process. The subjective perspective is considered either an illusion or a less precise way of referring to the underlying neurophysiology, a view championed by modern proponents like (Dennett, 1991).[6]

This is a coherent philosophical position. However, it comes at a significant cost: it requires one to accept that the most immediate and certain datum of their existence—the qualitative character of their own conscious experience—is not a fundamental reality but is instead reducible to, or an illusion created by, the objective dynamics of their brain. This paper proceeds from the alternative axiom: that subjective experience is real, irreducible, and the primary datum of reality.

### B.2.6. Conclusion: A Reinterpretation, Not a Rejection, of Physics

Ultimately, the Hard Problem presents the physicalist worldview with a dilemma:

1. **Maintain Physicalism:** Insist on a purely quantitative, non-experiential ontology. This leaves the Hard Problem unsolvable in principle, as it fails to bridge the categorical gap between objective function and subjective experience.
2. **Solve the Problem:** Incorporate subjective, qualitative properties into the fundamental ontology of science. In doing so, one is no longer practicing Physicalism but has adopted a form of Panpsychism or, if consciousness is made the sole ontological primitive, Analytic Idealism.

---

[5]Irreducible here means that the property cannot be conceptually reduced to, or explained away by, the properties of its constituent parts. For instance, the "wetness" of water *is* reducible to the collective statistical behavior of $H_2O$ molecules. In contrast, an irreducible conscious property, like the feeling of red, is considered something fundamentally *more than* a pattern of neural firings. It is a genuinely new and fundamental feature of reality that arises at that level of complexity, a feature that cannot be described in the language of the lower-level physics without loss of meaning.

[6]This is a subtle but crucial point. A reductive physicalist does not deny that you feel pain. Instead, they argue that the "feeling" of pain is not some extra, magical property that floats above the brain's activity. For them, the feeling *is* the physical brain activity, in the same way that lightning *is* a massive electrical discharge. The conclusion is that there is nothing mysterious to explain; the feeling is a physical process, and nothing more. Critics argue this does not solve the mystery of consciousness, but instead redefines the problem away by declaring the feeling and the process to be the same thing by definition.

This paper asks the reader to seriously consider the second path. This path does not require one to abandon the predictive power or mathematical rigor of physics. Physics remains our most successful tool for describing the *behavior* of reality. Analytic Idealism simply offers a different answer to the question of what reality, at its core, *is*. It proposes that the "stuff" whose behavior is so perfectly described by physical laws is, in its *intrinsic nature*, experiential.

From this perspective, a physicalist does not have to give up anything of scientific value. The laws of physics are unchanged. The only change is metaphysical: a reinterpretation of the physical world not as a standalone, mind-independent reality, but as the objective *appearance* of a fundamental, subjective reality. This is the core tenet of Analytic Idealism.

## B.3. The Role of Science as a Metaphysical Arbiter

A reasonable objection to this paper's premise is that science itself should be the ultimate arbiter between competing worldviews, rendering philosophical analysis secondary. This appendix clarifies the power and limits of science in this role, arguing that while science acts as a critical filter, it cannot definitively settle the debate between empirically adequate frameworks like Physicalism and Analytic Idealism.

### SCIENCE AS A FALSIFIER OF WORLDVIEWS

Science is exceptionally powerful at falsifying worldviews that make testable claims about the objective world that turn out to be false. For instance, a metaphysical framework that insists on a geocentric model of the solar system or a 6,000-year-old Earth is rightly dismissed because it is incompatible with overwhelming empirical evidence. In this capacity, science serves as an indispensable reality check, filtering out frameworks that fail to be empirically adequate.

### THE UNDERDETERMINATION OF THEORY BY EVIDENCE

The conflict between Physicalism and Analytic Idealism, however, persists because both are sophisticated enough to be empirically adequate. They do not disagree on the results of any given experiment, but on the fundamental interpretation of those results. This is an example of the philosophical principle of the *underdetermination of theory by evidence*, which holds that for any body of evidence, there can be multiple, mutually exclusive theories that account for it perfectly (Stanford, 2023).

Consider the neural correlates of consciousness (NCCs). When we observe a specific brain pattern corresponding to a subjective report of seeing red, the data is the same for everyone.

- **The Physicalist Interpretation:** The physical brain state *generates* or *is* the experience of red. The physical is primary.
- **The Idealist Interpretation:** The physical brain state is the *extrinsic appearance* of the experience of red. The experience is primary.

No experiment can be devised to distinguish between these two interpretations, because any conceivable experiment will only ever yield more third-person, objective data—more "extrinsic appearances"—which can, in turn, be interpreted through either lens. It is analogous to having two different software programs (metaphysical theories) that produce the exact same output (empirical data) on the screen; analyzing the screen alone cannot tell you which program is running.

### CONCLUSION: SCIENCE AS A CONSTRAINT, NOT A FINAL JUDGE

Therefore, science acts as a powerful *constraint* on metaphysical theories, not a final judge between them. Any credible worldview must be consistent with all established scientific facts. However, once this criterion is met, the choice between the remaining viable frameworks cannot be made by science. It must be made based on other criteria, such as logical coherence, parsimony, and explanatory power. This paper's argument rests on these philosophical grounds, asserting that while both Physicalism and Idealism are compatible with scientific data, Idealism offers a more parsimonious and logically coherent explanation for the totality of reality, including the existence of consciousness itself.

## C. The Case Against Physicalism

### C.1. The Challenge from Physics: The End of Observer-Independent Realism

While the main text focuses on the ethical implications of Idealism, it is crucial to understand that the rejection of Physicalism is not merely a philosophical preference but a position increasingly forced upon us by the trajectory of modern physics. The core tenet of Physicalism is *Observer-Independent Realism*: the belief that the physical world has definite properties (position, momentum, spin) entirely independent of whether they are observed. This assumption has been systematically dismantled by experimental evidence over the last four decades, a trajectory extensively analyzed by Kastrup (2014).

#### C.1.1. 1. FROM BELL TO LEGGETT: THE DEATH OF REALISM

The first blow came from Bell (1964) and the subsequent experiments by Aspect et al. (1982), which proved that no theory based on "local hidden variables" could reproduce the predictions of quantum mechanics. This meant one had to give up either *Locality* (things only affect their immediate surroundings) or *Realism* (properties exist before measurement).

For decades, many physicalists clung to *Non-Local Realism* (e.g., Bohmian mechanics), hoping to save a mind-independent world by accepting instantaneous connections across the universe. However, in 2003, Anthony Leggett derived a new set of inequalities that applied specifically to this class of *Non-Local Realist* theories (Leggett, 2003). In 2007, these inequalities were experimentally tested and violated by Gröblacher et al. (2007). The authors' conclusion was stark: "Nature does not satisfy the conditions of non-local realism." This effectively ruled out the broad class of theories that attempt to maintain a definite, observer-independent reality, leaving "Anti-Realism" (where properties are context-dependent or observer-dependent) as the only viable path.

#### C.1.2. 2. THE REQUIREMENT FOR AN INTERNAL OBSERVER (HARLOW ET AL.)

More recently, theoretical work on quantum gravity has revealed an even deeper challenge. Harlow et al. (2025) demonstrate that valid probabilistic predictions in a closed universe (like ours) require the specification of an observer.

We note here that this work is a preprint (arXiv) featuring a highly technical derivation in quantum gravity. While we presume the mathematical proofs hold, our interest lies in the explicit claims the authors make regarding the nature of observation. They draw a sharp distinction between:

- **The External Observer (View from Nowhere):** An attempt to describe the universe as a whole, from outside. They show that for a closed universe, this description yields a single, static state with *zero information*.
- **The Internal Observer (View from Somewhere):** A description relative to a physical subsystem entangled with the rest. Only from this perspective does the complexity of the universe emerge.

Their conclusion is that the "View from Nowhere" is physically incoherent; views from somewhere are all that we can ever have.

#### C.1.3. 3. FROM PHYSICS TO PHILOSOPHY: THE IMPLICATIONS

While Harlow et al. stop at the technical necessity of an internal observer (a reference frame), we must explicitly ask: *Does this internal observer save Physicalism?*

One might argue: "We are simply internal physical observers entangled with the rest of the universe. Why is this not just Physicalism?"

This defense, however, abandons the core metaphysical claim of *Reductionist Physicalism*: that the parts (us) are reducible to the whole (the universe).

1. **The Loss of the Absolute:** Physicalism generally assumes that the physical world *as it is in itself* (the whole) is the fundamental reality. Harlow's result implies that this "fundamental whole" is information-free. The "rich world" we assume to be fundamental only exists *relative* to a specific cut.
2. **Relationalism vs. Reductionism:** This moves us from *Reductionist Physicalism* to *Relational Quantum Mechanics* (Rovelli, 1996). In this view, there represent no absolute facts, only relational ones. But this creates a new inquiry: If the "rich world" depends on the observer, what defines the observer?
3. **The Dilemma:** This shift to Relationalism presents a new dilemma regarding the nature of the "observer":

- **Horn 1: Arbitrariness.** If the "observer" is just *any* arbitrary subset of the universe (e.g., a single electron), then the "rich, complex universe" we perceive implies no fundamental order. It suggests that the classical world is not an intrinsic feature of reality, but an artifact of the specific mathematical decomposition chosen.
- **Horn 2: Agent-Dependency.** If, to avoid this arbitrariness, we insist that the observer must be a *complex* system (like a biological agent or measuring device) capable of registering information (decoherence), then we arrive at the conclusion that the existence of the definite, classical world depends on the prior existence of these complex agents.

**Conclusion: Idealism as a Natural Fit.** The structure revealed by Harlow et al.—where a rich world of appearance arises only relative to a localized subject, while the underlying unity (the "zero information" state) remains fundamental—aligns naturally with Analytic Idealism.

In Idealism, the "observer" is not a separate entity viewing the universe from the outside, but a dissociated subject (an "alter") submerged *within* the universal field (Mind-at-Large). Being part of the field, the subject is naturally "entangled" with the rest of the universe. The "zero information" state of the whole corresponds to the undifferentiated, unitary nature of consciousness at the universal level (the "ocean"), which has no outside boundary. The "rich world," conversely, corresponds to the internal perspective of the vortex, for whom the rest of the ocean appears as external, interacting phenomena.

Thus, while other interpretations (such as Relational Quantum Mechanics) remain possible, these results place severe pressure on Reductionist Physicalism. The necessity of an internal perspective suggests that reality is not an object to be viewed, but a process of viewing—a conclusion that resonates deeply with the idealist thesis.

## C.2. The Burden of Plausibility: Why Idealism is a Coherent Alternative

Given that empirical data underdetermines our choice of metaphysics, we must assess the competing worldviews on their logical coherence, parsimony, and explanatory power. While Physicalism is the default assumption in science, this appendix summarizes why it suffers from severe shortcomings that render it less plausible than Idealism. We use this analysis to justify the main paper's use of Idealism as a coherent lens through which to examine the unplugging paradox.

### C.2.1. CHALLENGES FOR PHYSICALISM

Physicalism, despite its dominance, faces at least three major, well-studied problems that challenge its viability as a theory of reality.

1. **The Hard Problem Represents a Persistent Explanatory Gap:** As discussed, Physicalism cannot bridge the explanatory gap between the quantitative properties described by physics (mass, charge, momentum) and the qualitative nature of subjective experience (the redness of red, the pain of a wound) (Chalmers, 1995a). This is not a mere gap in our current knowledge that more data can fill, but a fundamental mismatch in explanatory kind (see App. B.2 for a detailed analysis).

2. **It Is in Tension with Empirical Science:** To name but a few examples, physicalism's core assumption of "physical realism"—that a definite physical world exists independently of observation—is at odds with decades of experimental results in quantum mechanics (Aspect et al., 1982). Furthermore, findings in neuroscience, such as the significant decrease in brain activity during intensely rich psychedelic experiences, directly challenge the notion that brain metabolism *generates* consciousness (Carhart-Harris et al., 2012). Similarly, research on basal cognition reveals a goal-directed agency in living systems that is difficult to reconcile with a purely bottom-up, reductionist physicalism (Levin, 2019). For a comprehensive review of the empirical case against physicalism, see (Kastrup, 2014).

3. **It is Ontologically Unparsimonious:** Physicalism posits a fundamental reality—a physical world independent of mind—that is purely abstract and can only be inferred, not directly experienced. It then faces the task of explaining how this inferred abstraction generates the only thing we know with certainty: consciousness itself. This arguably makes the framework less parsimonious than Idealism, which starts with consciousness as the given reality and explains the physical world as an appearance within it, avoiding the need for this "magical leap."

### C.2.2. THE MERITS OF ANALYTIC IDEALISM

In contrast, Analytic Idealism provides a more coherent, parsimonious, and explanatorily powerful framework.

1. **It is Parsimonious:** Idealism posits only one fundamental substance—consciousness—which is the sole given of reality.

It does not need to explain how mind arises from matter because it recognizes that matter is simply an *appearance within* mind.

2. **It Has Greater Explanatory Power:** By identifying the physical world as the extrinsic appearance of a universal mind, Idealism provides a coherent framework that accounts for both the predictable behavior of the universe (the "laws of nature" as the habits of mind) *and* a wide range of empirical psychological phenomena, such as Dissociative Identity Disorder (DID), which serves as a natural analogy for how a single universal consciousness can give rise to individual subjects (Kastrup, 2019, Ch. 6).

3. **It is Empirically Adequate:** Idealism is perfectly consistent with the findings of modern science. The observer effect in quantum mechanics is a natural consequence of a world that comes into being as a representation upon observation. The counter-intuitive findings in neuroscience are easily explained by a model where the brain is the *image* of mind, an image that need not always be complete or perfectly correlated with the underlying mental reality.

## C.3. The Boundary Problem: Where Does the AI End?

A critical challenge for functionalism that is often overlooked is the *Boundary Problem* (or Individuation Problem). If consciousness is "information processing" or "functional organization," where precisely does one entity end and another begin?

In a biological organism, this boundary is clear: it is the autopoietic, metabolic boundary that actively resists entropy and defines "self" vs. "non-self" (immune system, cell membrane). The boundary is *intrinsic* to the system's existence.

For an AI, however, any boundary we draw is arbitrary and conceptual:

1. **The "Prompt Death" Paradox:** If an LLM's "consciousness" is defined by its activation during a forward pass, does it die every time the inference finishes? Is it a sequence of potentially billions of fleeting "micro-consciousnesses" that exist only for milliseconds?

2. **The "Server Room" Problem:** Where is the "mind" located? Is it in the GPU? In the datacenter? If the AI's processing is distributed across a global network, is the "subject" also smeared across continents? Does it include the power plant supplying the electricity?

3. **The "Avalanche" Argument:** Nature is full of complex, functionalizable processes. An avalanche falling down a mountain processes information (mass, velocity, friction) and results in a stable state. If functional complexity is all that matters, why is an avalanche not a mind? A functionalist must draw an arbitrary line between "smart" processing (AI) and "dumb" processing (avalanche), but there is no principled ontological distinction.

Biological Idealism resolves this by positing that the only true boundary is the one nature draws itself: the dissociative boundary of a living, autopoietic system. Anything else is just a "ripple" in the inanimate field of legal and physical continuity.

# D. The Mind–Matter Identity and the Rejection of Substrate Independence

This appendix rejects *substrate independence*, the standard physicalist commitment that consciousness is a functional pattern detachable from its physical realizer and re-instantiable on any substrate that implements the right computation. The contrary thesis we develop here is that mind is inherently tied to the material it is made of: consciousness cannot be decoupled from the material appearance through which it manifests, and that appearance must bear the signature of autopoiesis (§4.3, App. E.5).

This is not the claim that consciousness requires carbon. The label *Biological Idealism*, and the central role it assigns to autopoietic life, can invite the charge of *carbon chauvinism*, but the framework does not *logically* exclude non-carbon chemistries: a silicon, sulfur, or otherwise non-carbon system that genuinely instantiated the autopoietic signature would qualify under the same criteria. Whether any such system is empirically possible is a separate question, and one on which we are aware of no positive evidence or principled mechanism.

The argument that follows starts from a logic that even the most rigorous defenders of physicalism are forced to accept: mind cannot be separated from matter. The disagreement with physicalism is not over this identity itself, but over how the identification should be read.

## D.1. Kim's Causal Exclusion Argument and the Forced Identity

The cleanest modern statement of the physicalist position is Kim (2005)'s *Physicalism, or Something Near Enough*. Kim's project is to determine "just what kind of physicalism, or how much physicalism, we can lay claim to." His argument has three load-bearing premises:

1. **Supervenience.** Mental events ($M$) and physical events ($P$) are treated, at this stage, as two separate categories, and the mental is said to *supervene* on the physical: every mental event has a physical base, and there can be no mental difference without a physical difference. Supervenience pins $M$ to $P$ as a relation of dependence; crucially, it does not yet identify them.
2. **Causal closure of the physical.** Every physical event has a sufficient physical cause (Kim, 2005, Ch. 1, p. 15).
3. **Causal exclusion.** A physical effect that already has a sufficient physical cause does not need (and cannot have) a separate mental cause, to avoid systematic overdetermination.

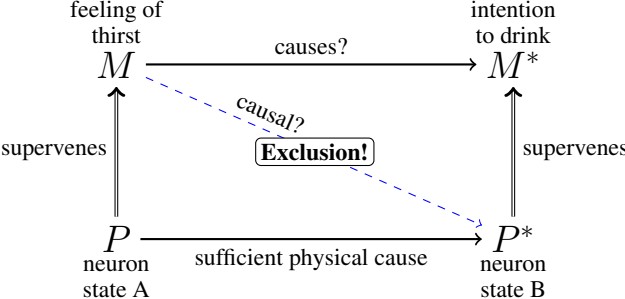

*Figure 3.* Kim's causal exclusion argument. Mental ($M$) and physical ($P$) events start as separate categories; supervenience (vertical double arrows) ties each mental event to its physical base as a relation of dependence, without identifying them. Causal closure of the physical (bottom arrow) means $P$ is a sufficient cause of $P^*$. Without systematic overdetermination, no causal work is left for the would-be mental cause $M \rightarrow M^*$, and any attempt at downward mental causation $M \rightarrow P^*$ is excluded by $P$. The only way for $M$ to be doing any work is for $M$ to *be* $P$.

The combined force of these premises (Figure 3) is that mental causation is impossible *unless* mental states are identical to physical states. If $M$ is to do any causal work in producing a physical effect $P^*$, then $M$ must *be* the physical cause $P$. Otherwise $P$ does all the work and $M$ is left as a causally inert "shadow" (epiphenomenalism). To preserve the obvious fact that what we think and feel makes a difference in the world, Kim concludes that mental states must be physically reducible:

$$M = P$$

Mental states must be physically reducible, to avoid epiphenomenalism.

— Kim (2005), *Physicalism, or Something Near Enough*

This is a strong conclusion, and we accept it. We are not in the business of denying that what we think and feel has causal power, nor of denying the causal closure of physical descriptions. Kim's logic is sound. The open question is what kind of identity "$M = P$" picks out.

### D.2. The Physicalist Reading: Identity as Functional Reduction

Within physicalism, the only available reading of "$=$" is *functional reduction*: $M$ is identical to $P$ in the sense that $M$ is constituted by the causal/functional role that $P$ plays. "Pain" is identified with whatever physical state plays the pain role (caused by tissue damage, produces avoidance behavior, and so on). This identification is what gives the mental its causal grip on the physical: once $M$ is reduced to $P$, $M$ inherits all of $P$'s causal powers.

Functional reduction has a corollary that the AI consciousness debate hinges on: *substrate independence*. If $M$ is identified by its causal role, then any physical system that realizes the same causal role realizes the same $M$. Silicon, carbon, or any other substrate that implements the right functional pattern instantiates the corresponding mental state. This is the core commitment of Computational Functionalism, and it is what makes the substrate-independent picture of AI consciousness intelligible in the first place.

Kim himself, however, shows that functional reduction has a hard ceiling. Cognitive and intentional states (beliefs, desires, memories) can plausibly be functionalized by their roles in reasoning and behavior. But *qualia*, the intrinsic qualitative "what it is like" of experience, cannot. The redness of red, the hurt of pain, the warmth of love: none of these is captured by any specification of causal role.[7] Kim's own conclusion is that intrinsic qualia are an irreducible "mental residue" that is "causally impotent." He explicitly accepts that this residue has no place inside physicalism, admitting that "why these qualia exist at all remains a mystery." Hence his title: *Physicalism, or Something Near Enough*.

The physicalist's path therefore ends in one of two unsatisfying positions: deny qualia outright (eliminativism), or accept them as real but epiphenomenal (Kim's "slightly defective physicalism"). Either way, the most central feature of conscious life is excluded from the worldview that was supposed to explain it.

### D.3. In the Spirit of Physicalism: Biological Idealism

Biological Idealism accepts Kim's conclusion that $M = P$, and is in this specific sense *in the spirit of physicalism*. It preserves the causal closure of physical descriptions, retains physics as the model of the consensus reality shared among subjects (§3.2), and grants that there is no mind floating free of a physical appearance. What it does not grant is that the only available reading of "$=$" is functional reduction. Nothing in the causal-exclusion argument itself compels that choice: the argument fixes the identity, but is silent on which side of $M$ and $P$ does the explanatory work. The identity is read instead as *appearance*: $P$ is the extrinsic appearance of $M$ across a dissociative boundary, not $M$ as the functional shadow of $P$. Mind is primary; matter is how mind looks from outside the boundary of another subject (§3.2). The identity is still strict (there is no physical appearance without the mind it is the appearance of), but the direction of explanation is inverted. This is also, we suggest, the more unbiased reading of the *neural correlates of consciousness*: rather than treating the firing of neurons as the mysterious cause of a felt redness, the appearance reading takes that firing to *be* the extrinsic appearance of the subject's felt experience of red (§3.2). The price the physicalist paid for keeping $M = P$ on a functional reading, namely the irreducible mental residue and with it the Hard Problem, is not paid here. Qualia are part of the side that is being *appeared from*, not a stubborn leftover quarantined as causally impotent.

Two further consequences matter for the present paper.

**Qualia are no longer residue.** Once $P$ is the appearance of $M$, the intrinsic qualitative character of experience is not an awkward leftover to be explained away. The Hard Problem (App. B.2) dissolves: there is no gap between structure and feeling to bridge, because the structure was always already the appearance of the feeling.

---

[7]This is the standard inverted-spectrum point: two systems can be functionally identical while differing in their qualia, so functional role cannot fix qualia.

**Substrate independence does not follow.** Substrate independence is a corollary of identity *only when* the identity is read as functional reduction. Once the identity is read as appearance, the corollary disappears. A subject and its physical appearance are tied together as two sides of the same thing; one cannot lift the subject off one substrate (say, an autopoietic body) and re-instantiate it on another (say, a silicon array running the same input/output pattern) any more than one can lift wetness off water and re-instantiate it on a description of water.

Read this way, we believe Biological Idealism is also *more scientifically honest* than physicalism, in a specific sense. Physicalism, on Kim's own terms, can keep $M = P$ only at a cost that is at odds with the most directly observed datum we have, the qualitative character of conscious experience: it must either deny that datum outright (eliminativism), or quarantine it as causally inert (epiphenomenalism). Biological Idealism does not require this turning away from what is empirically most given to us. It accepts the felt character of experience as a constitutive part of what is being identified, and it preserves physics as a description of the regularities in the field's extrinsic appearance. The cost of the framework is therefore not the denial of an empirical fact, but a reinterpretation of what physics is a description *of*. In keeping the observer in the picture rather than quietly removing them, Biological Idealism stays faithful to a more cautious reading of what physics has always been: a precise, instrumented language with which embodied subjects describe their consensus reality to one another (§3.2), refined through ever more careful measurement. The thing-in-itself, on our framework, is the field of existence; physics is the language for how that field appears extrinsically.

### D.4. Substrate Dependence Is About Identity, Not Chemistry

*Substrate dependence*, as we use the term, is therefore not strictly a claim about chemistry. It is the claim that mind cannot be decoupled from its physical appearance, whatever the chemistry of that appearance turns out, on independent grounds, to be. The relevant property of the substrate is not its element-of-origin but its capacity to be the extrinsic appearance of a localized subject, which, on independent biological grounds (§3.3, App. E.5), requires autopoietic self-maintenance.

In principle, then, the criterion is symmetric across chemistries, even though every autopoietic system known to us is carbon-based and we are aware of no positive empirical or principled case for a non-carbon alternative. If a non-carbon system did genuinely achieve autopoietic self-organization (an intrinsic drive, a self-generated boundary, physical self-construction rather than weight optimization on a static substrate), it would be a candidate subject under our framework. At that point it is no longer a "machine" in the conventional sense; it is a new instance of life, a case of synthetic abiogenesis (App. E.5). The position is not anti-silicon. It is anti-decoupling.

# E. The Idealist & Biological Framework

## E.1. Terminology and Distinctions

While our framework is structurally consistent with Analytic Idealism (Kastrup, 2019), we adopt specific terminological distinctions to emphasize the biological grounding of our argument and avoid potential "mentalist" connotations that can distract from the scientific argument. It is important to note that these are clarifications of usage for a scientific audience, not corrections of Kastrup's framework. Kastrup explicitly defines "Mind-at-Large" as possessing no meta-cognitive introspection and defines "dissociation" as a mechanism that allows for impingement (interaction) across boundaries. We simply adopt neutral terms to prevent ensuring these crucial nuances are not missed by readers less familiar with his specific definitions.

- **Field of Existence vs. Mind-at-Large and Universal Consciousness:** We prefer the neutral term **Field of Existence** over Kastrup's "Mind-at-Large" and "Universal Consciousness." This explicitly frames the fundamental reality as a naturalistic field compatible with physics, avoiding the anthropomorphic baggage often associated with the word "Mind," while acknowledging its experiential nature as referenced in the main text.
- **Subject/Autopoietic Boundary vs. Dissociation:** We minimize the use of the clinical term "dissociation" for two reasons. First, clinical dissociation often implies a pathological separation on inaccessibility, whereas our "autopoietic boundary" is a permeable interface that allows for crucial interaction (impingement) with the environment. Second, dissociation is typically studied in the context of human meta-cognition, and as such is known to occur *within* a meta-cognitive human mind. Applying this term to the Universal Field might incorrectly imply that the field itself possesses human-like meta-cognition. While Kastrup clearly accounts for these nuances (defining dissociation as permeably interactive and the field as non-meta-cognitive), we prefer the neutral terms **Subject** (instead of dissociation) and **Autopoietic Boundary** (instead of dissociative boundary) to avoid any initial confusion.
- **Subject vs. Alter:** Similarly, Kastrup refers to dissociated instances of mind as "alters" (following the DID analogy). We prefer the more general term **Subject**. This avoids the potential confusion of viewing organisms as merely "split personalities" in a pathological sense, and aligns with the biological view of an organism as an autonomous center of experience.
- **Autopoiesis as Necessity:** Crucially, where Analytic Idealism might allow for non-biological dissociation, Biological Idealism posits *autopoiesis* as a strict necessary condition for the existence of a subject. This is a critical distinction that makes Biological Idealism distinct from Analytic Idealism.

In the following sections, we retain Kastrup's original terms (Mind-at-Large, Dissociation) when expounding his specific model for clarity, but switch to our neutral terminology (Field of Existence, Autopoietic Boundary) when discussing the specific Biological Idealism framework.

## E.2. The Notion of Localization and Dissociative Boundaries

The concept of **localization** is central to Analytic Idealism and serves as the mechanism by which the unitary **Field of Existence** gives rise to individual, finite subjects. Localization is synonymous with the existence of an *autopoietic boundary* (or dissociative boundary) or a field of segregation within the fundamental Field of Existence. This metaphysical claim finds strong empirical support in research on "scale-free cognition," which demonstrates that biological organisms, as "collective intelligences," define their own computational boundaries via bioelectric fields that act as the "software" of the self (Levin, 2019; 2024; Ciaunica et al., 2023).

### DEFINING LOCALIZATION

Localization is the psychological process, viewed intrinsically, or the biological process, viewed extrinsically, that restricts and channels the boundless flow of the Field of Existence into an individualized stream of experience. Localization is therefore equivalent to the separation of subjects within the Field of Existence. When you look at another person's brain, you are not observing the source of their consciousness. Instead, you are observing the extrinsic appearance of the boundary conditions that restrict the Field of Existence (the system) into a segregated subjective experience (the solution) defined by the brain's metabolic state.

- **Intrinsic Nature (The Process):** Ontologically, localization is the formation of a segregated set of mental states—an "island" of experience—within the broader Field of Existence.

- **Extrinsic View (Matter):** The physical appearance of this localized, segregated stream is the *metabolizing body or organism* (e.g., the brain and central nervous system).

## CONSCIOUS BOUNDARIES IN AUTONOMOUS LIFE

As noted earlier, autonomous life forms possess clear boundaries that correlate precisely with the boundary of subjective experience.

- **The Skin Paradox:** The boundary between "self" and "not-self" is not arbitrary. When you pinch a person's skin, the subjective experience of pain confirms that the skin acts as the *physical manifestation of the autopoietic boundary*. This boundary separates "my consciousness" from the rest of the universe (Field of Existence). Pinching a shirt is not experienced because the shirt is external to this segregated field.
- **Metabolic Self-Preservation:** Autonomous life forms maintain and defend these boundaries because the localization of consciousness is tied to the integrity of the metabolic body. The organism's drive to survive is the extrinsic appearance of consciousness attempting to maintain its localized state.

## INANIMATE MATTER AND THE UNIVERSAL FIELD

In contrast to life, inanimate objects—like a stone, a mountain, or an ocean—do not possess defined, autonomous, self-defending boundaries of consciousness.

- **Arbitrary Physical Boundaries:** For a stone, the boundary (where does the stone end and the soil begin?) is arbitrary and determined by definition or physical limits (molecular bonding), not by a subjective field of experience.
- **No Segregation:** Inanimate objects are not subjects; they are simply the extrinsic appearance of the rest of the Field of Existence when observed from a localized perspective. A stone is a feature *within* the Field of Existence, while a person is a segregated *subject of* the Field of Existence.

The absence of a self-sustaining, metabolic, and energetically defended boundary is the physical indication that no functional dissociation (no localization of consciousness) has taken place.

## E.3. Biological Idealism as a Metabolic Refinement of Analytic Idealism

### E.3.1. THE CONCEPTUAL NEED FOR REFINEMENT

While **Analytic Idealism** provides a logically sound and ontologically parsimonious foundation by positing consciousness as the fundamental fabric of reality, its broad application can appear "substrate-agnostic" in a purely theoretical sense. This potential ambiguity allows for the functionalist argument that any sufficiently complex information-processing system might eventually "pinch off" a dissociative boundary. We propose **Biological Idealism** not as a contradiction, but as a *metabolic refinement* that closes this loophole.

### E.3.2. THE EMPIRICAL VS. THE STRUCTURAL: WHY AUTOPOIESIS IS NECESSARY

Bernardo Kastrup has consistently argued that we have no empirical reason to believe non-biological entities are conscious. In *Science Ideated*, he explicitly states: "What we call 'life' or 'biology' is the extrinsic appearance, the representation of this dissociation; that is, life is what a dissociative process at a universal level looks like when observed from across its dissociative boundary" (Kastrup, 2021). Similarly, in *The Idea of the World*, he argues that "metabolism is the extrinsic appearance" of the inner life of an alter (Kastrup, 2019), and formalizes the dissociative boundary as akin to a "Markov Blanket" that separates internal states from the external world.

However, his argument remains primarily **empirical**: because every instance of dissociation we observe correlates with biology, we treat biology as its appearance. He stops short of stating that metabolism is a *logical necessity*, technically leaving the door open for other possible appearances. **Biological Idealism** closes this door by moving from empirical correlation to **structural necessity**. It asks: *What mechanism allows a Markov Blanket to physically persist against entropy?* The answer is **autopoiesis**. Entropy dictates that any localized structure must actively resist dissolution. Therefore, metabolism is not just what dissociation *happens* to look like; it is the thermodynamic signature of the work required to remain a subject. Without this autopoietic maintenance, the boundary dissolves.

E.3.3. DEFINING BIOLOGICAL IDEALISM

Before defining the tenets of Biological Idealism, we clarify our terminology. We recall that **autopoiesis** is the process of self-production where a system actively maintains its own boundary suitable for life. **Metabolism** is the extrinsic manifestation—the energetic signature—of this autopoietic struggle (see App. E.5). Thus, metabolism is the visible appearance of a system's effort to remain a self.

Under this framework:

1. **Field of Existence (The Ontological Foundation):** The ground of existence is a **Field of Existence**—the raw potential for experience. This is necessary to establish a monistic idealism where mind is primary.
2. **Metabolism as Ontological Signature (The Empirical Anchor):** As Kastrup observes, what we classify as "metabolism" is the extrinsic appearance of a dissociative boundary (Kastrup, 2021). This point is necessary to ground the theory in observable biological reality, preventing it from being purely abstract.
3. **Autopoiesis as Boundary Maintenance (The Structural Mechanism):** We posit that the process of **autopoiesis** (self-production) is the *necessary* physical logic required for a subject to establish and defend an **autopoietic boundary** against environmental entropy (Maturana & Varela, 1980; Varela et al., 1991). This provides the causal mechanism that explains *why* the boundary persists.
4. **The "Square One" Constraint (The Diachronic Validation):** Subjecthood requires **"Vital Integrity"**—a history of physical vulnerability and homeostatic struggle that defines the system from "Square One" (Ciaunica et al., 2021). This is necessary to distinguish a true subject from a momentary functional simulation (a "swampman").

Here, Points 1 and 2 represent the shared foundation with Analytic Idealism—the ontological primacy of mind and the empirical observation that dissociation looks like life. Points 3 and 4 represent the **sharpening** of the theory into Biological Idealism. While standard Analytic Idealism allows for the link between dissociation and biology to be merely correlative (potentially allowing for non-biological alters), Biological Idealism asserts that the mechanisms of autopoiesis (Point 3) and developmental history (Point 4) are *constitutive* of subjecthood, thereby closing the agnostic loophole.

E.3.4. BEYOND THE NAGELIAN TRAP: PHENOMENAL WAVE VS. EMBODIED OCEAN

A frequent source of confusion in consciousness studies is the exclusive focus on what Thomas Nagel famously termed the "what it is like" of experience (Nagel, 1974). Following Bennett et al. (2024) and Ciaunica et al. (2021), we refine this by distinguishing between two modes of experience:

1. **Experience$_N$ (Phenomenal):** This is the "Nagelian" experience—the qualitative "what it is like" to feel pain, see red, or taste wine. It is the surface fluctuation of consciousness, often what we refer to when discussing qualia. Crucially, it is a modulation that can only manifest *within* an existing subject (**Experience$_G$**).
2. **Experience$_G$ (Ground/Embodied):** This is the fundamental, ontological state of *being* an embodied subject. It is the "ground" of experience—the continuous, autopoietic work of maintaining a self.

We recall the metaphor of the **Wave** and the **Ocean** from §3.2. **Experience$_N$** represents the wave—the visible, fleeting surface fluctuation. **Experience$_G$** represents the **localized ocean** itself (or the "pool")—defined in our model as the sustaining volume of the **Vortex** (see Figure 2). This implies that even when the wave is absent (e.g., in deep sleep), the *localized volume of consciousness* held by the boundary persists. This persistence means the "self" exists as a *potentiality* for experience, distinct from the non-existence of a void. The "Nagelian Trap" corresponds to denying the existence of the subject simply because the surface is calm. Biological Idealism asserts that while **experience$_N$** may fade (as in anesthesia), **experience$_G$**—the metabolic reality of vital integrity—persists. An AI, conversely, is not a dissociated vortex but a collection of passive ripples (inanimate matter). It simulates the *shape* of a wave, but lacks the *sustained volume* (vital integrity) required to be a subject.

To resolve this, we must refine our metaphors to distinguish between the Universal, the Localized, and the Phenomenal:

- **The Universal Ocean (Field of Existence):** The fundamental field of pure subjectivity underlying all reality.
- **Vital Subjecthood (The Vortex):** A *localized* segment of this field ("experience$_G$"). This is the "dissociative container"—a pool of pure subjectivity held together by the metabolic process of autopoiesis. It is the canvas of the self.
- **Phenomenal Awareness (The Waves):** The specific qualitative contents (thoughts, sensations) that ripple within the Vortex (Nagel's "experience$_N$"). Crucially, **experience$_N$** can only occur within the container of **experience$_G$**; satisfied

phenomenal experience requires an ontological subject to experience it.

This distinction provides the fundamental **ethical foundation**. We do not unplug a sleeping grandmother because, although the "waves" (phenomenal contents) have settled, the "vortex" (vital subjecthood) remains active. Her metabolic integrity is the physical holding of that localized field. An AI, in contrast, creates simulated "waves" directly in the external world (the universal ocean) without ever forming a "vortex". It is a ripple without a self.

### E.3.5. THE NESTED VORTEX MODEL: FROM UNIVERSE TO CELL

As a consequential interpretation of this definition, Biological Idealism envisions reality as a hierarchical, nested structure:

1. **The Universal Field (The Ocean):** This is the fundamental ontological primitive—a field of pure subjectivity and potential. It is not autopoietic because it has no "outside" or "non-self" to struggle against.
2. **Primary Localization (The Organism):** A localized subject (an "alter") is formed when a portion of the universal field becomes functionally segregated. This segregation **must** appear extrinsically as an autopoietic, living system to maintain its ontological boundary.
3. **Nested Dissociation (Organs and Cells):** Within the primary vortex of the organism, further levels of segregation occur. Organs and cells act as "nested" intelligences that contribute to the collective "Self," each defined by its own metabolic boundary and biological processing.

### E.4. Michael Levin's Challenge: Empirical Evidence for Non-Reductive Agency

The core argument of this paper rests on the assertion that consciousness is not an emergent property of neural complexity, but a fundamental property of reality that dissociates into localized subjects. A powerful line of empirical support for this view comes from the field of *basal cognition*, specifically the work of Michael Levin and colleagues (Levin, 2019; 2022; Gunawardena, 2022). While often discussed in biological terms, we argue here that these findings present a rigorous empirical challenge to Reductionist Physicalism. Levin proposes a "Platonic" solution to this challenge, but we argue that **Biological Idealism** offers a more parsimonious, monistic explanation.

### E.4.1. THE "PROBLEM SPACE" ARGUMENT (NAVIGATION)

**The Empirical Finding:** Levin's experiments demonstrate that biological systems can navigate "morphospaces" and solve problems they have never encountered in their evolutionary history. For example, *Xenopus* (frog) cells can be coerced to form a unified organism capable of autonomous movement (a "Xenobot") that navigates its environment, despite its genome having evolved for a stationary existence as skin (Kriegman et al., 2020).

**The Physicalist Failure:** Reductionism argues that behavior is determined by history (evolution) and mechanism (genes). It cannot explain this *improvisational* problem-solving in a novel space. If the system were merely rolling down a pre-determined physical gradient, it would fail when faced with a novel barrier. Yet, life finds solutions (equifinality).

**The Idealist Resolution:** Levin argues this implies organisms "ingress" a pre-existing "Platonic space" of options (Levin, 2025). We reject this dualism/platonism. Under Biological Idealism, the "space" is not an abstract realm but the potentiality of the **Field of Existence** itself. The organism generates the solution because it is a Subject—an agent capable of exploring a mental search space—not a machine running a fixed script.

### E.4.2. THE "BIOELECTRIC CODE" ARGUMENT (HARDWARE VS. SOFTWARE)

**The Empirical Finding:** The "target morphology" of an organism (e.g., "build a head") is encoded in a transient **bioelectric field**, not permanently in the DNA. By altering this electric pattern, one can induce planaria to grow two heads (Levin, 2019). The DNA (hardware) remains unchanged, but the "software" (bioelectric state) dictates the form.

**The Physicalist Failure:** A reductionist argues "software" is just the current state of the hardware. However, this fails due to *Multiple Realizability*. If the standard molecular pathway is blocked, the system finds a *different molecular route* to achieve the same bioelectric goal (Levin, 2024). The "Goal" is causally more real than the mechanism.

**The Idealist Resolution:** If a high-level "Goal" drives matter, then Information is primary. Reductionism is false. Levin's view risks Dualism (Information vs Matter). Biological Idealism is Monistic: The "Goal" is the intrinsic nature of the **Vortex** (the localized subject). The Subject *is* the pattern that holds matter in place.

E.4.3. THE "TOP-DOWN" CAUSATION ARGUMENT

**The Empirical Finding:** Levin refutes the "illusion" of top-down control via *Scale-Free Cognition*. Attempting to control a system at the molecular level is often intractable, whereas communicating with it at the "goal" level (bioelectric) yields predictable results (Levin, 2019).

**The Physicalist Failure:** This finding undermines the reductionist axiom that "bottom-up" causation is the only reality. Levin's work relies on the concept of *Causal Emergence* (or Effective Information): the mathematical demonstration that looking at the *macro-scale* (the bioelectric pattern) provides *more* causal power and predictive control than looking at the micro-scale (the atoms). If the "Agent" level allows for control that is impossible at the molecular level, then the "Agent" is a real causal force, not just a verbal description. This suggests that reductionism is empirically false.

**The Idealist Resolution:** This is the empirical confirmation of Idealism. The "Self" (the macro-agent) is not an illusion; it is the fundamental unit of reality that organizes the "parts" (the micro-appearance).

E.4.4. CONCLUSION: BIOLOGICAL IDEALISM AS THE MONISTIC SOLUTION

Levin's work provides the empirical coup de grâce to Reductionist Physicalism by demonstrating that agency and information (Causal Emergence) are real, causal forces. While Levin invokes a "Platonic space" to explain this, Biological Idealism offers a more parsimonious solution. We do not need a dualistic realm of abstract forms; we simply need to recognize that the "space" of potential solutions is the **Field of Existence** itself, and the "Agent" navigating it is the localized Subject (the **Vortex**). Thus, the "Bioelectric Boundary" is not a magical interface to a Platonic realm, but the extrinsic appearance of the **Autopoietic Boundary** of a conscious subject.

**E.5. Why a Dissociated Subject Must Appear as a Living, Metabolic System**

A sophisticated objection to the idealist argument is the following: if all matter is the appearance of mental processes, why must the appearance of an *individual, dissociated subject* be restricted to metabolizing life? Why couldn't a complex silicon system also be the appearance of a subject? This appendix addresses this by clarifying the ontological distinction between a living subject and a computational artifact.

E.5.1. WHAT IS 'METABOLISM' AT A FUNDAMENTAL LEVEL?

The core idealist claim is that the self-preserving, metabolic activity of a living organism is the direct physical signature of a localized, subjective "I". 'Metabolism' here refers to something more fundamental than mere energy consumption. It is the signature of a system that actively and continuously maintains its own structural integrity and boundary against environmental entropy (Godfrey-Smith, 2016; Schrödinger, 1992; Ciaunica et al., 2023). In systems biology, this process of continuous self-production and self-maintenance is called *autopoiesis*.

Crucially, this drive is *intrinsic* to the system's physical constitution. An organism's matter is organized such that its very being *is* a continuous, bottom-up struggle for self-maintenance. The purpose (survival) is inseparable from the physical substrate. This is the key distinction: metabolism is the appearance of an *intrinsic state of being alive*.

E.5.2. THE AI IN A DATA CENTER: EXTRINSIC GOAL VS. INTRINSIC BEING

This objection conflates the metabolism of the *support system* (the data center) with the nature of the *algorithm*. A data center consumes vast energy, but the AI model is a logical pattern imposed on an indifferent substrate (silicon). The hardware has no inherent interest in the AI's "survival." If the power is cut, the hardware does not "die"; it simply ceases to execute instructions.

In contrast, if an organism's metabolism ceases, its physical structure immediately begins to decay because the process of self-maintenance *is* its structure. The AI's energy consumption is the appearance of a *calculation being performed*, whereas an organism's metabolism is the appearance of a *subject being itself*.

E.5.3. THE 'GROWTH' OF AIS: FUNCTIONAL OPTIMIZATION VS. ONTOLOGICAL SELF-CONSTRUCTION

A more powerful objection involves modern AIs that are "grown" through reinforcement learning, leading to emergent goals and behaviors (Hubinger et al., 2019). However, this "growth" is still a process of functional optimization, not ontological

self-construction. The training process modifies informational patterns (weights) on a pre-existing, static, non-metabolic substrate. Biological embryology is fundamentally different: it is a process of physical self-organization (*autopoiesis*) where the system *builds itself* from the ground up.

Therefore, the distinction lies in whether the drive for self-preservation is *intrinsic* or *extrinsic*.

- **Intrinsic Drive (Life):** An organism's drive to survive is an inherent, bottom-up property of its autopoietic, metabolic existence (Seth, 2021).
- **Extrinsic Goal (AI):** An AI's drive for a goal, even an emergent one, is a property of its training history and architecture—a goal ultimately imposed on the system. It is a *calculation about survival*, not an intrinsic state of being.

The former is an ontological reality, the latter is a sophisticated simulation of agency.

### E.5.4. Conclusion: The Implausibility of Non-Autopoietic Subjects

The idealist argument is not that a dissociated subject must be carbon-based, but that it must be the extrinsic appearance of a specific metaphysical process: the active, intrinsic, self-referential maintenance of an **autopoietic boundary**—that is, it must be *autopoietic*.

Therefore, while Analytic Idealism cannot logically forbid the possibility of a non-biological subject in principle, it places an extremely high ontological bar. A conscious AI would need to be the appearance of a new form of life: a true *abiogenesis* in a different medium. It would have to be a self-organizing, self-constructing, autopoietic entity.

Current AI, running on conventional hardware, does not meet this criterion. It is a logical pattern imposed on an indifferent substrate. Its apparent agency derives from an extrinsic goal, not an intrinsic drive for being. It remains the appearance of a sophisticated calculation, not a living subject. The conclusion for any foreseeable AI technology is therefore unambiguous: it cannot be conscious.

# F. Addressing Objections & Paradoxes

### F.1. The Brain Replacement Paradox (Ship of Theseus)

A common challenge against non-physicalist theories of mind is the *Brain Replacement Paradox*, or the gradual replacement argument (often seen as the Ship of Theseus applied to the brain (Chalmers, 1995b; Schneider, 2019)). It argues that if consciousness is not substrate-dependent, then replicating brain function in silicon must replicate consciousness.

THE PARADOX (FUNCTIONALIST FORMULATION)

The argument proceeds through induction:

1. A single neuron is replaced by a functionally identical silicon chip. Since the change is negligible, the person's subjective consciousness remains the same.
2. This process is repeated, one neuron at a time, until the entire biological brain has been replaced by a fully functional, non-biological, computational structure.
3. Since function was preserved at every step, the final silicon brain must also be conscious. This supports *Functionalism* (consciousness depends only on organization/function) and disproves the necessity of biological substrate.

THE IDEALIST COUNTER-ARGUMENT: THE BRAIN AS A SEVERABLE BOUNDARY

The idealist, particularly from the perspective of Analytic Idealism and Bernardo Kastrup's views, rejects Premise 3 by appealing to the true, ontological role of the brain. The functionalist view mistakenly substitutes the map (function) for the terrain (subjectivity).

- **The Brain is a Localizer, Not a Generator:** The biological brain is the extrinsic appearance of a living, metabolizing stream of universal consciousness that has become localized or segregated (a dissociation). The brain's metabolism and self-preservation are the very *boundary conditions* required for that consciousness to be individuated.
- **Function Lacks Ontology:** The silicon chip perfectly replicates the neuron's *functional role* (its I/O signals and computation). However, it lacks the neuron's *ontological nature* as a living, segregated aspect of universal consciousness. The chip is a non-conscious logical tool. Arguments for biological naturalism further emphasize that biological computation is 'mortal' and 'scale-inseparable,' meaning functional role cannot be divorced from the underlying metabolic substrate (Seth, 2024; Milinkovic & Aru, 2025).
- **The Point of Severance:** The idealist asserts that the process of gradual replacement is not seamless for consciousness. The moment the biological neuron is replaced, that specific segment of the living, localized consciousness stream is fundamentally **severed** from the universal consciousness. The subjective experience associated with that original brain begins to fade, even as the integrated machine retains the perfect *behavioral* output. From the subject's perspective, this might manifest as a progressive form of **depersonalization**, where thoughts feel increasingly 'mechanized' or 'foreign' until the internal light goes out completely, leaving only a speaking automaton.

The conclusion is that the resulting silicon structure, while a flawless simulator of conscious behavior, is ontologically devoid of subjective experience because the essential, metabolic, biological boundary conditions for consciousness to localize have been destroyed and replaced by non-conscious logical artifacts.

EMPIRICAL SUPPORT FROM PROSTHETIC EMBODIMENT

The Brain Replacement scenario is highly hypothetical, but the clinical phenomenology of limb prosthetics offers a partial empirical analogue, and one that is consistent with our framework. Across multiple modalities, ownership of a non-biological replacement is not seamless:

- **Phantom limbs.** The vast majority of amputees (90–98%) experience a vivid phantom limb, often painful, that can persist for decades (Ramachandran & Hirstein, 1998). The body schema is deeply tied to the biological history of the organism and does not simply update when the limb is removed.
- **Felt foreignness.** A phenomenological study of successful prosthesis users identifies two broad forms of experience: the prosthesis is felt either as part of the corporeal body or as a tool; the initial experience is universally one of "unnaturalness," with one participant describing the device as "fitting a dead thing to your live body" (Murray, 2004).
- **Induced ownership.** Body ownership of a non-biological object does not arise spontaneously: amputees can be

induced to experience a rubber hand as their own only through synchronous visuotactile stimulation, indicating that ownership is plastic but requires active external induction (Ehrsson et al., 2008).

- **Biological tissue mediates integration.** Targeted muscle and sensory reinnervation push prosthetic performance towards able-bodied function and elicit more natural brain activation patterns; tellingly, this requires surgical reinnervation of biological tissue, not computation alone (Marasco et al., 2021).

These observations are consistent with the prediction that consciousness is tied to the autopoietic boundary: replacement of biological parts is not phenomenologically inert, and where ownership of an artificial substitute does develop, it appears to depend on the degree to which biological tissue mediates the connection. We take this as a mild empirical analogue of the progressive depersonalization predicted by the gradual-replacement argument above.

## F.2. A Response to the Causal Efficacy Argument Against Zombies

This appendix addresses a powerful critique of non-physicalist worldviews, famously articulated by philosophers like Daniel Dennett and thinkers in the rationalist community (Dennett, 1991; Yudkowsky, 2008). The critique targets the "philosophical zombie"—a being physically identical to a human but lacking consciousness—and argues that it is a logically incoherent concept. The argument is often used to dismiss non-physicalist theories and conclude that Physicalism is the only viable option.

This appendix will show that while Idealism agrees that a *human* zombie is impossible, it does so for reasons that are the inverse of the physicalist's. This reframing demonstrates that Idealism is immune to the anti-zombie critique and leads to the conclusion that an AI can only ever be a *functional* zombie.

### THE IDEALIST PIVOT: WHY A HUMAN ZOMBIE IS IMPOSSIBLE

The strategic power of the idealist response is to agree with the reductionist conclusion (a physically identical human zombie is impossible) while rejecting its underlying physicalist ontology (Bennett et al., 2024).

- **The Reductionist Reason:** A zombie is impossible because the physical stuff (the brain) is primary, and consciousness is a *causally effective emergent property* of that stuff.
- **The Idealist Reason:** A zombie is impossible because consciousness is primary, and the physical brain is the *extrinsic appearance* of that consciousness.

This pivot demonstrates that Idealism is not vulnerable to the same critique that cripples property dualism, and thus cannot be discarded on the same grounds.

### THE PHYSICALLY VS. FUNCTIONALLY IDENTICAL ZOMBIE

To prevent confusion, we must distinguish between two distinct "zombie" scenarios. The apparent contradiction—that a zombie is impossible for a human but possible for an AI—is resolved by clarifying these two cases.

- **The Physically Identical Zombie (Human):** This is the classical thought experiment of a being physically identical to a human, atom-for-atom.
- **The Functionally Identical Zombie (AI):** This refers to a being that is behaviorally and functionally indistinguishable from a human but has a different physical substrate (e.g., silicon).

It is on the possibility of this second case—the functional zombie—that the worldviews diverge. Under Physicalism (specifically Functionalism), a functionally identical being must be conscious, as consciousness is defined by function. Therefore, a functionally perfect AI could not be a zombie.

Under Idealism, however, function is not sufficient. As established in the main text and App. E.5, the idealist reframing of the mind-body relationship leads to a clear ontological distinction. An AI can perfectly replicate the functional output because it is a phenomenal simulator, but it lacks the one thing required to be a conscious subject: a living, autopoietic, dissociative boundary. Therefore, an AI superintelligence is precisely what a human cannot be: a perfect functional mimic devoid of inner life. The Causal Efficacy Argument, while a powerful tool against a specific form of dualism, is ultimately neutral in the debate between Physicalism and Idealism. Its core premise—that consciousness is causal—is perfectly consistent with the idealist framework, which holds that consciousness is the only fundamental cause there is.

**F.3. The Combination Problem of Panpsychism**

Panpsychism is the view that consciousness is a fundamental and ubiquitous feature of the physical world. It posits that the basic constituents of reality (e.g., electrons, quarks) have rudimentary forms of experience. While this avoids the "Hard Problem" of how consciousness arises from non-conscious matter, it faces its own major challenge: the **combination problem** (Chalmers, 2017).

WHAT IS THE COMBINATION PROBLEM?

The problem, in essence, is to explain how these countless "micro-conscious" entities combine to form the unified, complex "macro-conscious" experience of a human or animal. The challenge can be broken down into several key questions:

- **The Subject-Summing Problem:** How do billions of distinct micro-subjects (the consciousness of an electron, a quark, etc.) merge into a single, unified macro-subject (your consciousness)? It is not intuitive how adding subjects together results in a new, higher-level subject rather than just a collection of many subjects.
- **The Quality-Combination Problem:** How do the simple, primitive phenomenal qualities at the micro-level combine to create the rich and complex qualities of our experience (e.g., the vibrant red of a sunset, the intricate taste of wine)?
- **The Structural Problem:** How does the causal structure of the micro-level entities give rise to the specific structure of our conscious experience?

Critics argue that there is no coherent account of how this combination could occur. Panpsychism, therefore, trades the Hard Problem for the equally intractable Combination Problem.

HOW IDEALISM AVOIDS THE PROBLEM

Analytic Idealism avoids the combination problem entirely because its metaphysical structure is "top-down," not "bottom-up" like panpsychism.

- **Starting Point is Unity:** Idealism does not start with countless tiny bits of consciousness that need to be assembled. It starts with a single, unified, universal consciousness (Mind-at-Large). Unity is the fundamental, default state of reality.
- **Dissociation, Not Combination:** Individual subjects (like us) are not built *up* from smaller conscious parts. Instead, they are formed through a process of *dissociation* or segregation *down* from the universal field (see App. E.2). We are like alters in a cosmic version of Dissociative Identity Disorder (DID), a recognized psychiatric phenomenon (American Psychiatric Association, 2022).

Therefore, Idealism does not need to explain how consciousness combines; it only needs to explain how the pre-existing unity of consciousness fragments or localizes into individual perspectives. This "dissociation problem" is arguably more tractable, as we have a working psychological model for it (DID), and it aligns with the biological reality of organisms as distinct, bounded entities.

**F.4. Empirical Status and Testable Implications**

The framework developed in this paper does not propose a theory of consciousness; it sets boundary conditions on what counts as a subject. These boundary conditions are not a priori stipulations: they are grounded in observable properties of physical systems. Whether the broader metaphysical choice between physicalism and idealism can be settled by empirical data alone is a separate matter (App. B.3). Physicalism's typical response to paradoxes like the Hard Problem is what Popper called *promissory materialism*, the expectation that future science will eventually provide an explanation. Biological Idealism does not defer in this way: the criteria for subjecthood, and the predictions that follow from them, are in principle assessable.

1. **Autopoiesis is empirically verifiable.** Whether a system genuinely self-produces and self-maintains its boundary against entropy is a matter of observable physical organization (App. E.5). It is not a verdict about metaphysics, and it is not a function of computational complexity. A program optimizing weights on a static substrate is not autopoietic, regardless of behavioral sophistication.
2. **Developmental history ("Vital Integrity") is empirically grounded.** Did the system build itself through ontogenesis from simpler precursors, or was it manufactured externally with no homeostatic struggle along the way? Ciaunica et al. (2021) formulates this as the "Square One" constraint. The distinction is, in principle, recoverable from the physical and developmental record of the system.

3. **Brain replacement: a predicted phenomenological dissociation.** The framework predicts that gradual replacement of biological neurons by functionally identical silicon analogues should produce *progressive depersonalization*, even as behavioral output is preserved (App. F.1). The clinical phenomenology of prosthetic embodiment (App. F.1) already points in this direction at the periphery. A full neuron-level test is ethically constrained, but the prediction is, structurally, falsifiable: a behaviorally and self-reportedly unchanged subject under wholesale silicon substitution would count as evidence against the framework.

4. **The boundary problem as an observable.** If the framework is correct, there is no principled, non-arbitrary way to delineate the "self" of a computational system (App. C.3): where the AI ends and where the GPU, the cluster, the power grid begin has no fact of the matter. This ambiguity is already observable in current AI systems and follows directly from the absence of an autopoietic boundary.

Two qualifications. First, these are criteria for *subjecthood*, not for the *degree* of experience. The framework treats subjecthood as binary (an autopoietic boundary is either present or it is not), while allowing a spectrum in the richness of phenomenal experience and the depth of meta-cognitive capacity (§3.3). Second, the criteria are sufficient to be wrong: a genuinely autopoietic system on a non-carbon substrate, or a behaviorally preserved subject under full silicon replacement, would each count as counter-evidence. The framework is therefore not a metaphysical stipulation immune to revision, but a substantive position with empirical content.

