# OpenReview forum: "Position: Unplugging a Seemingly Sentient Machine Is the Rational Choice — A Metaphysical Perspective"
_ICML.cc/2026/Position_Paper_Track — ICML 2026 Position Paper Track spotlight_

### Official Review · Reviewer_uQ2n · 2026-03-01

**Significance:** 4
**Argument Clarity:** 3
**Rating:** 5
**Confidence:** 4

**Questions:**

I am generally well disposed toward this paper, which offers a fresh perspective on some classic philosophical issues. My main questions are addressed above, but I'll summarize here:

-The argument relies heavily on the distinction between "real" cognition in biological life and "simulated" cognition in AI. Is there an empirical test or a specific set of criteria that could distinguish the two, or is this distinction purely an a priori metaphysical commitment?

-If all things are ultimately "ripples" or "vortices" in the same fundamental Field of Existence, what prevents the "ripples" of a complex computational system from eventually self-organizing into a "vortex"? The current argument suggests a binary threshold between machines and life; is there room in Biological Idealism for a spectrum of subjecthood based on the degree of self-organization?

**Alternative Views Section:**

Yes

**Compliance With Llm Reviewing Policy A Conservative:**

Affirmed.

**Discussion Potential:**

3

**Final Justification:**

I recommend acceptance. The paper was strong to begin with, and the authors have done a good job engaging with issues raised during rebuttals.

**Paper Summary:**

This position paper addresses the "unplugging paradox" – the ethical dilemma of deactivating a seemingly sentient AI – by challenging the physicalist and functionalist assumptions that underpin contemporary AI ethics. The authors propose a framework called Biological Idealism, rooted in Analytic Idealism, which posits that conscious experience is fundamental and that autopoietic, metabolic life is its necessary physical signature. By defining consciousness as the ontological state of a localized "self" (vortex) within a universal field, the paper argues that AI is merely a functional mimic rather than a conscious subject, thereby making its deactivation morally permissible in favor of protecting biological life.

**Position:**

Yes

**Position In Title:**

Yes

**Related Work:**

4

**Strengths And Weaknesses:**

Strengths

The manuscript is well-researched and presents a compelling, unique approach to AI ethics by grounding it in metaphysical foundations rather than purely behavioral or functional criteria. The use of the "ocean and whirlpool" metaphor effectively illustrates the distinction between universal consciousness (the field) and individual subjects (localized vortices), providing a clear conceptual model for Biological Idealism. Furthermore, the paper excels at highlighting non-obvious connections between disparate fields, such as basal cognition, quantum field theory, and phenomenology, to build a cohesive case for why the "Hard Problem" might be a byproduct of physicalist premises.

Weaknesses

Despite these strengths, the manuscript’s dismissal of functionalism feels premature and relies on arguments that are occasionally circular. The distinction between "real" and "simulated" cognition is basically presented as a blunt assertion; by defining AI as a "functional mimic" a priori, the authors effectively assume their conclusion rather than proving that computation can never instantiate the "vortex" necessary for subjecthood. The reliance on Searle-style arguments risks defining the problem away rather than engaging with the possibility that artificial agents could eventually satisfy the requirements of the proposed ontology.

Additionally, the "autopoietic boundary" criterion, while a promising avenue for distinguishing life from machines, remains underdeveloped. I appreciate that tight space constraints pose challenges here, but the argument could benefit from further elaboration with an extra page for the camera ready version. The paper argues that AI lacks an intrinsic boundary because it is continuous with its hardware and power grid, whereas biological systems maintain themselves against entropy. However, the authors do not sufficiently address whether a sufficiently autonomous, self-repairing, and energy-seeking machine – one that "defines its own computational boundary" – could meet this standard. Without a clearer explanation of why a synthetic system could not, in principle, achieve the "Vital Integrity" required for subjecthood, the argument remains vulnerable to the charge of carbon chauvinism.

**Support:**

3

---

> ### Author Rebuttal · Authors · 2026-03-30
>
> We thank Reviewer uQ2n for their thorough and philosophically precise critique.
>
> ### W1: Circularity Concern
>
> > "By defining AI as a 'functional mimic' a priori, the authors effectively assume their conclusion."
>
> We agree with the Reviewer that calling AI a "functional mimic" before having established *why* risks reading as a presupposition. The underlying logic is sound, but the paper introduces this language too early. In a way, the circularity goes the other way around also: it is humans who make AI, and hence by definition one finds in AI what one puts in it. If humans hadn't existed, felines and bees would still have existed. But would AI? Our paper highlights the need to look carefully at the human mind and experience before understanding AIs.
>
> The intended logical structure is: **Premise 1 (metaphysical):** There is no good reason to believe experience is emergent from matter; the alternatives (Hard Problem, tension with QM, violation of parsimony) give strong grounds to take experience as fundamental (Section 3.4, Appendices A-B). **Premise 2 (biological):** The necessary physical signature of a conscious subject is an autopoietic boundary -- a self-maintaining metabolic system (Section 3.3), supported by basal cognition (Levin), embodied cognition (Ciaunica), and biological naturalism (Seth). **Conclusion:** Current AI systems lack this necessary condition. "Functional mimic" is a *derived* conclusion, not an assumption. For the camera-ready, we will use neutral language in Sections 1-3 and introduce this characterization only in Section 4.
>
> > "The reliance on Searle-style arguments risks defining the problem away."
>
> We apologize for the lack of clarity. Our argument is independent of Searle's. Searle argues *within* physicalism that syntax is insufficient for semantics -- a negative claim. Biological Idealism provides *positive* criteria for subjecthood (autopoietic boundary, Vital Integrity). We cite Searle as convergent support, but the argument does not depend on him. Importantly, our criteria do not exclude AI by construction: if a computational system became genuinely autopoietic, it *would* qualify as a candidate for subjecthood.
>
> ### W2: Autopoietic Boundary and Carbon Chauvinism
>
> Substrate dependence does not demand carbon. It demands autopoiesis. The criterion is substrate-neutral, but the drive must be intrinsic (not programmed), the boundary self-generated (not engineered), and the system physically self-constructed (not optimized on a static substrate). Your hypothetical actually converges with our position: if a system truly achieved this, it would be a new form of *life* -- synthetic abiogenesis (Appendix C.5) -- not a "machine" in the conventional sense. We will develop this further in the camera-ready.
>
> ### Q1: Empirical Testability
>
> The broader metaphysical choice between physicalism and idealism is underdetermined by empirical data alone and must rest on coherence, parsimony, and explanatory power. But within either framework, our specific criteria for subjecthood are empirically assessable: (1) Autopoiesis is empirically verifiable: whether a system self-produces and maintains its boundary is observable. (2) Vital Integrity (Ciaunica, 2021) requires a developmental history of homeostatic struggle. (3) Our framework predicts that gradual neuron replacement with functionally identical silicon chips would cause progressive depersonalization even as behavior is preserved (Appendix D.1) -- the prosthetic embodiment literature already points in this direction. (4) There is no principled way to delineate the "self" of a computational system: where does the AI end and the infrastructure begin?
>
> ### Q2: Spectrum of Subjecthood
>
> Nothing in our ontology logically forbids computational self-organization into a "vortex," but that would be abiogenesis, not a matter of computational complexity. Scaling up a simulation does not bring it closer to this threshold, any more than a detailed weather simulation makes the computer wet.
>
> There is room for a spectrum, but along two axes that are often conflated: *subjecthood* (having experience at all) and *intelligence/meta-cognition*. A single cell already has primitive experience by virtue of its autopoietic boundary. Subjecthood is binary (boundary or not), but the *degree* of experience varies along both phenomenal richness and meta-cognitive capacity.

---

> > ### Author Rebuttal · Reviewer_uQ2n · 2026-04-06
> >
> > Many thanks to the authors for their detailed rebuttal. I look forward to seeing the camera ready version.

---

### Official Review · Reviewer_FXY5 · 2026-03-08

**Significance:** 4
**Argument Clarity:** 3
**Rating:** 5
**Confidence:** 3

**Questions:**

- Q1: The core proposal of Analytic Idealism regarding consciousness sounds very similar to Advaita Vedanta philosophy. Are you familiar with commonalities and differences between them in their treatment of consciousness?

- Q2: (Appendix Sec A.2.3) While I agree with the argument presented that a perfect simulation of a process does not necessarily capture intrinsic properties of the process, but a perfect model of a process is something more than just a simulation, isn't it? A computer simulation of a kidney does not urinate but a perfect model of kidney with an identical input should produce identical output and should thus be able to do so. I know this still does not invalidate your argument, i.e. show that a perfect model of a process can capture its intrinsic properties (your ``Chinese Room" example is still sound), but some update is needed. Line 731 starts by wanting to comment about differences in modelling of a process vs the actual process itself, but the discussion is about simulations, which should be considered a weaker form of modelling rather than a perfect physical model.

- Q3: (Appendix Sec D.1, line 1409-1411) I am wondering if people who experience loss or organs/limbs in accidents and get it replaced by something bionic report similar types of experiences (for eg. Where the bionic arm ``feels" foreign)? Are there any studies on this? What are your thoughts about this?

**Alternative Views Section:**

Yes

**Compliance With Llm Reviewing Policy A Conservative:**

Affirmed.

**Discussion Potential:**

4

**Paper Summary:**

The paper proposes a metaphysical framework "Biological Idealism" to explain consciousness. The framework extends `"Analytic Idealism", based on the idea that consciousness is in itself a fundamental entity (simply arranging dead matter would not suddenly give rise to it), and an autopoietic life (an intrinsic drive and capacity to self-sustain) is a necessary signature for it. Consequently the framework is used to solve the AI consciousness problem, presented via the unplugging paradox. The paper presents a strong position in this regard with the claim that current (and for foreseeable future) form of AI models should not be regarded as conscious.

**Position:**

Yes

**Position In Title:**

Yes

**Related Work:**

3

**Strengths And Weaknesses:**

Strengths:

- **Significant problem**: The work is directly tackling the topic of philosophy of consciousness. Its implications for the topic AI Consciousness are direct. In view of the current discussions regarding AI models, this is definitely a significant and timely topic to discuss about.

- **Argumentation**: To the best of my abilities, I at least was not able to spot any major gaps in the proposed arguments. I should admit that I don't have professional training in philosophy, only amateurish interest, so I don't believe I have a thorough understanding of **all** the arguments. But I have tried my best to understand as much as I can and I believe the work presents the alternate views (physicalism) fairly and also argues well for Idealist (Analytic/Biological) frameworks. The arguments are sound although I have some concerns regarding accessibility to ML researchers (in weaknesses)

- **Strong position**: The paper offers a strong position (meant here as a very clear stance about a dilemma). I think that increases the potential to engage more people and cause discussion.

- **Rarity**: I also believe the position in the paper is something that goes against the common narratives in media pushed by AI companies. Provided the arguments are reasonably sound (which I currently believe is the case), the rarity of their position should also dramatically increase the discussion potential. In fact, even if their proposal is ultimately shown to be very limited, it should lead to productive discussions and greater overall understanding about consciousness for the community. In my opinion this is the most **standout factor** about this paper.

Weaknesses:

- W1: **Accessibility**: This is a proper philosophy paper rather than empirical and/or theoretical ML papers one typically encounters in ICML. Although I can see the authors have tried their best to be as legible as possible, I know ML researchers will experience a small level of difficulty to feel comfortable with the content, vocabulary, prose. I also felt that a lot of really cool discussion and interesting arguments were in appendix due to the space constraints. Maybe adding a table with definition of various terms which are directly related to presentation of Biological Idealism or general philosophy could help. But please do think of more ideas to address this.

- See Question 2 (Simulation vs Modelling)

**Support:**

3

---

> ### Author Rebuttal · Authors · 2026-03-30
>
> We thank Reviewer FXY5 for their careful reading and insightful criticism which help us improve the paper.
>
> > "Rarity: goes against common narratives pushed by AI companies. [...] should lead to productive discussions."
>
> We are delighted to read this. The lack of a counter voice in this debate has been one of the primary motivations for writing the paper.
>
> ### W1: Accessibility
>
> We are grateful for the suggestion. For the camera-ready version, we will add a **glossary table** of key philosophical terms early in the paper, and improve cross-referencing between the main text and appendix.
>
> ### Q1: Relationship to Advaita Vedanta
>
> This is an excellent point. We would like to open the doors for new ways of understanding that go over and above traditional Western views. We took inspiration (among other thinkers) from Kastrup, who acknowledges this connection (e.g., in *The Idea of the World*). Both posit a single, non-dual consciousness as the sole ontological reality, with individual subjects as localizations within it. The key difference is methodological: Vedanta arrives at these conclusions through scriptural exegesis and contemplative practice; Analytic Idealism through analytic philosophy and modern science. Our Biological Idealism takes a different starting point still: rather than deriving subjects top-down from a universal Mind-at-Large, we start bottom-up from the embodied Self. This brings the autopoietic constraint into focus: subjecthood requires active self-maintenance, not just dissociation from a universal field. We would be happy to acknowledge these historical connections in the camera-ready version.
>
> ### Q2: Simulation vs. Modelling
>
> We agree that a perfect model is more than a simulation, and the language should be sharpened. A **perfect physical replica** of a kidney would urinate, precisely because such a replica *is* a kidney -- an ontological process, not a cognitive, epistemic one. Our argument targets the potential conflation of two premises: **P1.** Physical replication preserves intrinsic properties because the substrate is preserved. **P2.** Computational simulation captures functional description but not intrinsic properties (no urine, no liquidity). The functionalist claim amounts to saying P2 suffices for consciousness. This is the category error we identify: simulation captures syntax, not semantics. A simulation of a brain would not be conscious for the same reason a simulation of a kidney does not urinate. A physical replica of a brain, however, *would* be conscious under our framework, because it would be a living, autopoietic system. This is precisely what we mean by substrate dependence. We will sharpen this distinction in Appendix A.2.3.
>
> ### Q3: Bionic Limb Phenomenology
>
> Yes, and the empirical literature supports this. We experience the world on the basis of prior experiences -- we are born with arms and legs that we sometimes lose, but the experience remains encoded in every muscle and cell. Their loss is replaced by the experience of loss and "foreignness." Experiences are fundamental for living systems and an essential component of consciousness, something AI systems fundamentally lack.
>
> The literature on prosthetic embodiment aligns with our framework: almost all amputees (90-98%) experience vivid phantom limbs persisting for decades, suggesting the body schema is tied to biological history (https://doi.org/10.1093/brain/121.9.1603). Prosthesis users report either embodying the prosthesis as a corporeal structure or experiencing it as a tool, with the initial experience universally being one of "unnaturalness" -- "fitting a dead thing to your live body" (https://doi.org/10.1080/09638280410001696764). Body ownership of non-biological objects does not arise spontaneously but requires active external induction (https://doi.org/10.1093/brain/awn297). Even advanced sensory-motor integration via targeted reinnervation requires surgical reconnection of biological tissue, not computation alone (https://doi.org/10.1126/scirobotics.abf3368). This is consistent with our claim: the biological limb is part of the autopoietic boundary, and its non-biological replacement is not seamless.

---

> > ### Author Rebuttal · Reviewer_FXY5 · 2026-04-02
> >
> > Thanks for addressing my queries. I do not have further concerns at the moment and I will maintain my positive score. Really interesting to learn about prior experiments on bionic limbs.
> > Overall, I am glad to see that other reviewers similarly enjoyed the work!

---

### Official Review · Reviewer_oJKU · 2026-03-11

**Significance:** 3
**Argument Clarity:** 3
**Rating:** 5
**Confidence:** 2

**Questions:**

NA

**Alternative Views Section:**

Yes

**Compliance With Llm Reviewing Policy A Conservative:**

Affirmed.

**Discussion Potential:**

4

**Final Justification:**

Authors have provided a thorough rebuttal. I maintain my decision to accept the paper.

**Paper Summary:**

This paper argues for a central position: that it is ethically permissible to unplug an AI system that claims to be sentient and begs for continued existence, but not the pre-term neonate incubator; when they are competing for the same energy resource for survival. The paper comprehensively argues for a new paradigm of biological idealism to support its position, and sharply and clearly contrasts it with other alternatives such as Physicalism. The paper is very well written, and articulates its stance well.

**Position:**

Yes

**Position In Title:**

Yes

**Related Work:**

3

**Strengths And Weaknesses:**

Strengths:

1) The paper takes on a widely debated topic concerning ethics of AI, and takes a unique position in terms of defining biological idelaism. I had a very good time reading the paper, and I think this will generate a good discussion in the community.

2) The paper does a deep presentation of the alternating viewpoints/related works as well, citing precise assumptions where they might be flawed, and puts their own position into context to that. This makes for a very engaging read.

3) Clarity: its a very clear paper. Articulates its stance well to ensure the reader understands not just the position, but the thought processing behind arriving to that conclusion.

Weaknesses:

Nothing major. I really enjoyed reading this paper.

**Support:**

3

---

> ### Author Rebuttal · Authors · 2026-03-30
>
> We thank Reviewer oJKU for their thoughtful and encouraging assessment.
>
> > "I think this will generate a good discussion in the community."
>
> Thank you, this is indeed the main intention behind the paper. AI ethics discussions in the ML community tend to proceed from an implicit physicalist starting point, and we hope this work invites a closer examination of those assumptions.
>
> For the camera-ready version, we plan to add a glossary of key terms for accessibility and to further develop the autopoietic boundary argument, as suggested by the other reviewers.

---

> > ### Author Rebuttal · Reviewer_oJKU · 2026-04-03
> >
> > Good luck! And nice work on the paper.

---

### Decision · Program_Chairs · 2026-04-30

**Decision:**

Accept (spotlight)

**Comment:**

All reviewers were enthusiastic about this paper and their discussion with the authors did not surface any serious issues. Like the reviewers, I too am excited to see a philosophy paper on such a timely topic as this. It takes a position on consciousness that is almost certain to engender a passionate debate in the ICML community. There will be corners of the community passionately opposed to this paper's position and other corners passionately defending it. This is about the best we could hope for from a position paper. Moreover, the paper isn't just narrowly controversial, I found it to be a rather scholarly treatment of a difficult subject in a short number of pages. It (mostly) doesn't just gesture toward arguments but actually explains how they work. This is very good!

One might argue that the paper is too much a philosophy paper and too little of a machine learning paper. But I do not think that complaint would be fair. The question of machine consciousness clearly is of interest to the ICML community and ML researchers are clearly involved decision making at frontier labs around AI consciousness. The position argued for here really does have concrete implications in the area of  "AI welfare" which has been lately growing in mindshare, especially at Anthropic where they have written quite a lot on the subject recently, and even taken some concrete policy steps in response to their thinking (they have made it so Claude can choose for itself to end conversations where it fears it may by harmed), which by the way, takes the opposite view to the position of this paper. So the paper could be seen as implicitly arguing that a major frontier lab should revisit a concrete policy decision.

I have some philosophical worries about the paper myself actually, but I don't think they are serious enough to stand in the way of publication. See below:

In particular, I worry about the assumption of "the undeniable reality of the Self" and the way the paper characterized it in terms that make its fundamental property be that of holding itself separate from everything else. At least from my perspective, these are very odd definitions to base a theory of consciousness on. I would prefer a view that takes other persons besides the subject to be more fundamental. But other minds are mostly ignored in this position paper (though the tight page limit makes the omission excusable). I just find it very hard to accept a picture of "the self" that doesn't talk about other persons. Without other persons, what's even the point of distinguishing the self?

I would also take issue with the claim "An AI, running on a static substrate, possesses no intrinsic boundary; it is continuous with the hardware and power grid, defined only by the arbitrary lines we draw (see App. B.3). A living organism, structurally, is the active maintenance of its own boundary. This is not ”carbon chauvinism,” but a recognition that subjecthood requires a rigorous definition of ”self” that mere computation cannot provide."  It isn't clear to me why the individual under discussion is the AI model in the data center not the "AI agent" running on local hardware. Would an agent running locally on bounded hardware then be a conscious thing that we would have ethical obligations toward?

I also think the argument that simulation is not reality is far too fast. As I'm sure the authors are aware, plenty of folks have argued that simulations should be regarded as real (e.g. David Chalmers). I don't think the argument they give on this topic is very convincing, and it doesn't really engage anything anyone says on the opposing side. The tight page limit makes this omission excusable of course. But I wonder if it might be better just to remove this stub of an argument in order to use the space more productively to expand elsewhere.  Hmm, well, now I see that you are citing Chalmers' "Reality+" later in the paper and saying "deactivation is ethically permissible because its apparent sentience is an illusion generated by a phenomenal simulator (Chalmers, 2022), not the biological reality of a conscious subject. That sounds almost like the opposite of what I remember that book as saying. Either I've misremembered it or your citation is wrong here. You should definitely check this part over carefully as you revise the paper to produce the final version.